# Model, sample, and epoch-wise descents: exact solution of gradient flow in the random feature model

**Antoine Bodin**   **&**   **Nicolas Macris**

Communication Theory Laboratory,
School of Computer and Communication Sciences,
Ecole Polytechnique Fédérale de Lausanne
`antoine.bodin@epfl.ch`,
`nicolas.macris@epfl.ch`

## Abstract

Recent evidence has shown the existence of a so-called double-descent and even triple-descent behavior for the generalization error of deep-learning models. This important phenomenon commonly appears in implemented neural network architectures, and also seems to emerge in epoch-wise curves during the training process. A recent line of research has highlighted that random matrix tools can be used to obtain precise analytical asymptotics of the generalization (and training) errors of the random feature model. In this contribution, we analyze the *whole temporal behavior* of the generalization and training errors under gradient flow for the random feature model. We show that in the asymptotic limit of large system size the *full time-evolution* path of both errors can be calculated analytically. This allows us to observe how the double and triple descents develop over time, if and when early stopping is an option, and also observe time-wise descent structures. Our techniques are based on Cauchy complex integral representations of the errors together with recent random matrix methods based on linear pencils.

## 1 Introduction

Deep learning models have vastly increased in terms of number of parameters in the architecture and data sample sizes with recent applications using unprecedented numbers with as much as 175 billions parameters trained over billions of tokens [1]. Such massive amounts of data and growing training budgets have spurred research seeking empirical power laws to scale model sizes appropriately with available resources [2], and nowadays it is common wisdom among practitioners that "larger models are better". This ongoing trend has been challenging classical statistical modeling where it is thought that increasing the number of parameters past an interpolation threshold (at which the training error vanishes while the test error usually increases) is doomed to over-fit the data [3]. We refer to [4] for a recent extensive discussion on this contradictory state of affairs. Progress towards rationalizing this situation came from a series of papers [5, 6, 7, 8, 9, 10, 11] evidencing the existence of phases where increasing the number of parameters beyond the interpolation threshold can actually achieve good generalization, and the characteristic U curve of the bias-variance tradeoff is followed by a "descent" of the generalization error. This phenomenon has been called the *double descent* and was analytically corroborated in linear models [12, 13, 14, 15, 16] as well as random feature (RF) (or random feature regression) shallow network models [17, 18, 19, 20]. Many of these works provide rigorous proofs with precise asymptotic expressions of double descent curves. Further developments have brought forward rich phenomenology, for example, a triple-descent phenomenon [21] linked to the degree of non-linearity of the activation function. Further empirical evidence [22] has also shown that a similar effect occurs *while* training (ResNet18s on CIFAR10 trained using Adam) and has been

35th Conference on Neural Information Processing Systems (NeurIPS 2021).

called *epoch-wise double descent*. Moreover the authors of [22] extensively test various CIFAR data sets, architectures (CNNs, ResNets, Transformers) and optimizers (SGD, Adam) and classify their observations into three types of double descents: (i) model-wise double descent when the number of network parameters is varied; (ii) sample-wise double descent when the data set size is varied; and (iii) epoch-wise double descent which occurs while training. We wish to note that sample-wise double descent was derived long ago in precursor work on single layer perceptron networks [23, 24]. An important theoretical challenge is to unravel all these structures in a unified analytical way and understand how generalization error evolves in time.

In this contribution we achieve a detailed analytical analysis of the gradient flow dynamics of the RF model (or regression) in the high-dimensional asymptotic limit. The model was initially introduced in [25] as an approximation of kernel machines; more recently it has been recognized as an important playground for theoretical analysis of the model-wise double descent phenomenon, using tools from random matrix theory [17, 18, 26]. Following [17] we view the RF model as a 2-layer neural network with fixed-random-first-layer-weights and dynamical second layer learned weights. The data is given by $n$ training pairs constituted of $d$-dimensional input vectors and output given by a linear function with additive gaussian noise. The data is fed through $N$ neurons with a non-linear activation function and followed by one linear neuron whose weights we learn by gradient descent over a quadratic loss function. The high-dimensional asymptotic limit is defined as the regime $n, d, N \to +\infty$ while the ratios tend to finite values $\frac{N}{d} \to \psi$ and $\frac{n}{d} \to \phi$. As the training loss is convex one expects that the least-squares predictor (with Moore-Penrose inversion) gives the long time behavior of gradient descent. This has led to the calculation of highly non-trivial analytical algebraic expressions for training and generalization errors which describe (model-wise and sample-wise) double and triple descent curves [17, 21]. However, to the best of our knowledge, there is no complete analytical derivation of the whole time evolution of the two errors.

We analyze the gradient flow equations in the high-dimensional regime and deduce the whole time evolution of the training and generalization errors. Numerical simulations show that the gradient flow is an excellent approximation of gradient descent in the high-dimensional regime as long as the step size is small enough (see Fig. 2). Main contributions presented in detail in Sect. 3 comprise:

**a.** Results 3.1 and 3.2 give expressions of the time evolution of the errors in terms of *one and two-dimensional integrals over spectral densities whose Stieltjes transforms are given by a closed set of purely algebraic equations*. The expressions lend themselves to numerical computation as illustrated in Fig. 1 and more extensively in Sect. 3 and the supplementary material.

**b.** Model and sample-wise double descents develop after some definite time at the interpolation threshold and are preceded by a *dip or minimum* before the spike develops. This indicates that early stopping is beneficial for some parameter regimes. A similar behavior also occurs for the triple descent. (See Figs. 3, 4 and the 3D version Fig. 1).

**c.** We observe two kinds of epoch-wise "descent" structures. The first is a *double plateau* monotonously descending structure at widely different time scales in the largely overparameterized regime (see Fig. 3). The second is an *epoch-wise double descent* similar to the one found in [22]. In fact, as in [22], rather than a spike, this double descent appears to be an *elongated bump* over a wide time scale (see Fig. 5 and the 3D version Fig. 1).

Let us say a few words about the techniques used in this work. We first translate the gradient flow equations for the learned weights of the second layer into a set of integro-differential equations for generating functions, as in [27], involving the resolvent of a random matrix (constructed out of the fixed first layer weights, the data, and the non-linear activation). The solution of the integro-differential equations and the time evolution of the errors can then be expressed in terms of Cauchy complex integral representation which has the advantage to decouple the time dependence and static contributions involving traces of algebraic combinations of standard random matrices (see [28] for related methods). This is the content of propositions 2.0.1 and 2.0.2. With a natural concentration hypothesis in the high-dimensional regime, it remains to carry out averages over the static traces involving random matrices. This is resolved using traces of sub-blocks from the inverse of a larger nontrivial block-matrix, a so-called *linear pencil*. To the best of our knowledge linear pencils have been introduced in the machine learning community only recently in [29]. This theory is developed in the context of random matrix theory in [30, 31] and [32] using operator valued free-probability. A side-contribution in the SM is also an independent (but non-rigorous) derivation of a basic result of this theory (a fixed point equation) using a replica symmetric calculation from statistical physics.

The non-linearity of the activation function is addressed using the gaussian equivalence principle [33, 34, 29]. Finally, our analysis is not entirely mathematically controlled mainly due to the concentration hypothesis in Sect. 2.3 but comparison with simulations (see Fig. 2 and SM) confirms that the analytical results are exact.

In the conclusion we briefly discuss possible extensions of the present analysis and open problems among which is the comparison with a dynamical mean-field theory approach.

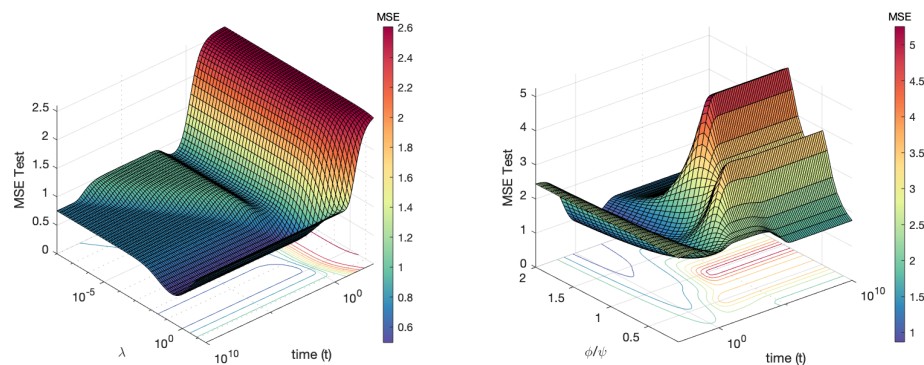

Figure 1: 3D plot of analytical test error evolution. See Figs. 5 and 4 (on the right) for parameter values.

## 2 Random feature model

### 2.1 Model description

**Generative model and neural network:** We consider the problem of learning a linear function $f_d(x) = d^{-\frac{1}{2}}\beta^T x$ with $x, \beta \in \mathbb{R}^d$ column vectors. The vector $x$ is interpreted as a random input and $\beta$ as a random *hidden* vector; both with distribution $\mathcal{N}(0, I_d)$, $I_d$ the $d \times d$ identity matrix. We assume having access to the hidden function through the noisy outputs $y = f_d(x) + \epsilon$ with additive gaussian noise $\epsilon \sim \mathcal{N}(0, s^2)$, $s \in \mathbb{R}_+$. We suppose that we have $n$ data-points $(x_i, y_i)_{1 \leq i \leq n}$. This data can be represented as the $n \times d$ matrix $X \in \mathbb{R}^{n \times d}$ where $x_i^T$ is the $i$-th row of $X$, and the column vector vector $Y \in \mathbb{R}^n$ with $i$-th entry $y_i$. Therefore, we have the matrix notation $Y = d^{-\frac{1}{2}}X\beta + \xi$ where $\xi \sim \mathcal{N}(0, s^2 I_n)$ and $I_n$ the $n \times n$ identity matrix.

We learn the data with a shallow 2-layer neural network. There are $N$ hidden neurons with weight (column) vectors $\theta_i \in \mathbb{R}^d$, $i = 1, \cdots, N$ each connected to the $d$ input neurons. Out of these we form the matrix (of the first layer connecting input and hidden neurons) $\Theta \in \mathbb{R}^{N \times d}$ where $\theta_i^T$ is the $i$-th row of $\Theta$ . Its entries are assumed independent and sampled through a standard gaussian distribution $\mathcal{N}(0, 1)$; they are not learned but fixed once for all. The data-points in $X$ are applied linearly to the parameters $\Theta$, and the output $Z \in \mathbb{R}^{n \times N}$ of the first layer is the *pointwise application* of an activation function $\sigma : \mathbb{R} \to \mathbb{R}$, $Z = \sigma(d^{-\frac{1}{2}}X\Theta^T)$. We use the notation $z_i^T$ to express the $i$-th row of $Z$. The second layer consists in a weight (column) vector $a_t \in \mathbb{R}^N$ to be learned, indexed by time $t \geq 0$, with components initially sampled at $t = 0$ i.i.d $\mathcal{N}(0, r^2)$, $r \in \mathbb{R}_+$. The prediction vector is expressed as $\hat{Y}_t = N^{-\frac{1}{2}}Za_t$.

We assume that the activation function belongs to $L^2(e^{-\frac{x^2}{2}}dx)$ with inner product denoted $\langle \, , \, \rangle$. It can be expanded on the basis of Hermite polynomials, so $\sigma \in \text{Span}\,((H_{e_k})_{k \geq 0})$, where $H_{e_k}(x) = (-1)^k e^{\frac{x^2}{2}} \frac{d^k}{dx^k} e^{-\frac{x^2}{2}}$ (so $H_{e_0}(x) = 1$, $H_{e_1}(x) = x$, $H_{e_2}(x) = x^2 - 1$, $H_{e_3}(x) = x^3 - 3x$, ...). Furthermore we take $\sigma$ centered with $\langle \sigma, H_{e_0} \rangle = 0$, and set $\mu = \langle \sigma, H_{e_1} \rangle$, $\nu^2 = \langle \sigma, \sigma \rangle - \mu^2$. For instance, $\sigma = \text{id}$ has $(\mu, \nu) = (1, 0)$ while $\sigma = \text{Relu} - \frac{1}{\sqrt{2\pi}}$ has $(\mu, \nu) = (\frac{1}{2}, \frac{1}{2}(1 - \frac{2}{\pi})^{1/2}) \simeq (0.5, 0.3)$. Finally, we recall that we are interested in the high dimensional regime where the parameters $N, n, d$ tend to infinity with the ratios $\frac{N}{d} \to \psi$ and $\frac{n}{d} \to \phi$.

**Training and test errors:** For a new input $x_0 \in \mathbb{R}^d$, we define the predictor $\hat{y}_t(x_0) = \frac{1}{\sqrt{N}} z(x_0)^T a_t$ where $z(x_0) = \sigma(\frac{1}{\sqrt{d}} \Theta x_0)$. We will further define the standard training and test errors with a penalization term $\lambda > 0$ and the quadratic loss:

$$\mathcal{H}^{\text{train}}(a) = \frac{1}{n} \left\| Y - \hat{Y} \right\|^2 + \frac{\lambda}{N} \|a\|^2, \quad \mathcal{H}^{\text{test}}(a) = \mathbb{E}_{x_0 \sim \mathcal{N}(0,1)} \left[ (y(x_0) - \hat{y}(x_0))^2 \right] \tag{1}$$

Note that because of the $\lambda$-penalization term, in this context, the training error can be above the test error in some configurations of parameters. Also, we will slightly abuse this notation throughout the paper by using $\mathcal{H}^{\text{train}}_t, \mathcal{H}^{\text{test}}_t$ to designate $\mathcal{H}^{\text{train}}(a_t), \mathcal{H}^{\text{test}}(a_t)$.

**Gradient flow:** Minimizing the training error of this shallow-network is equivalent to a standard Tikhonov regularization problem with a design matrix $Z$ for which the optimal weights are given by $a_\infty = (\frac{Z^T Z}{N} + \frac{n}{N} \lambda I_N)^{-1} \frac{Z^T}{\sqrt{N}} Y$. The errors generated by the predictors with weights $a_\infty$ have been analytically calculated in the high-dimensional regime in [17] and further analyzed in [21]. Here we study the *whole time evolution* of the gradient flow and thus introduce an additional time dimension in our model. Of course as $t \to +\infty$ one recovers the errors generated by the predictors with weights $a_\infty$. The output vector $a_t$ is updated through the ordinary differential equation $\frac{da_t}{dt} = -\eta \nabla_a \mathcal{H}^{\text{train}}(a_t)$ with a fixed learning rate parameter $\eta > 0$. As $\eta$ can be absorbed in the time parameter, from now on we consider without loss of generality that $\eta = \frac{n}{2}$. Setting $\delta = \lambda \frac{n}{N}$, we find that the gradient flow for $a_t$ is a first order linear matrix differential equation,

$$\frac{da_t}{dt} = -\left( \frac{Z^T Z}{N} + \delta I_N \right) a_t + \frac{Z^T Y}{\sqrt{N}}. \tag{2}$$

Recall the initial condition $a_0$ is a vector with i.i.d $\mathcal{N}(0, r^2)$ components.

## 2.2 Cauchy integral representations of the training and test errors

An important step of our analysis is the representation of $\mathcal{H}^{\text{train}}$ and $\mathcal{H}^{\text{test}}$ in terms of Cauchy contour integrals in the complex plane. To this end we decompose both errors in elementary contributions and derive contour integrals for each of them. Details are found in section 4 and the SM.

We begin with the test error which is more complicated. We have

$$\mathcal{H}^{\text{test}}_t = 1 + s^2 - 2\mu g(t) + \mu^2 h(t) + \nu^2 l(t) + o_d(1) \tag{3}$$

where $\lim_{d \to +\infty} o_d(1) = 0$ with high probability, and $g(t) = \frac{\beta^T}{\sqrt{d}} \frac{\Theta^T}{\sqrt{d}} \frac{a_t}{\sqrt{N}}$, $h(t) = \| \frac{\Theta^T}{\sqrt{d}} \frac{a_t}{\sqrt{N}} \|^2$, and $l(t) = \| \frac{a_t}{\sqrt{N}} \|^2$. To describe Cauchy's integral representation of the elementary functions $g, h, l$ we introduce the resolvent $R(z) = (\frac{Z^T Z}{N} - z I_N)^{-1}$ for all $z \in \mathbb{C} \setminus \text{Sp}(\frac{Z^T Z}{N})$.

**Proposition 2.0.1 (Test error)** *Let $\mathcal{R}_z$ be the functional acting on holomorphic functions $f : \mathbb{C} \setminus \text{Sp}(\frac{Z^T Z}{N}) \to \mathbb{C}$ as $\mathcal{R}_z\{f(z)\} = -\oint_\Gamma \frac{dz}{2\pi i} f(z)$ over a contour $\Gamma$ encircling the spectrum $\text{Sp}(\frac{Z^T Z}{N})$ in the counterclockwise direction. Similarly, let $\mathcal{R}_{x,y}$ be the functional acting on two-variable holomorphic functions $f : (\mathbb{C} \setminus \text{Sp}(\frac{Z^T Z}{N}))^2 \to \mathbb{C}$ as $\mathcal{R}_{x,y}\{f(x,y)\} = \oint_\Gamma \oint_\Gamma \frac{dx}{2\pi i} \frac{dy}{2\pi i} f(x,y)$. Let $G_t(z) = \frac{\beta^T}{\sqrt{d}} \frac{\Theta^T}{\sqrt{d}} R(z) \frac{a_t}{\sqrt{N}}$ and $K(z) = \frac{\beta^T}{\sqrt{d}} \frac{\Theta^T}{\sqrt{d}} R(z) \frac{Z^T Y}{N}$. We have for all $t \geq 0$*

$$g(t) = \mathcal{R}_z \left\{ e^{-t(z+\delta)} G_0(z) + \frac{1 - e^{-t(z+\delta)}}{z + \delta} K(z) \right\}. \tag{4}$$

*Let $L_t(z) = \frac{a_t^T}{\sqrt{N}} R(z) \frac{a_t}{\sqrt{N}}$ and $U_t(z) = \frac{Y^T Z}{N} R(z) \frac{a_t}{\sqrt{N}}$ and $V(z) = \frac{Y^T Z}{N} R(z) \frac{Z^T Y}{N}$. For all $t \geq 0$*

$$l(t) = \mathcal{R}_z \left\{ e^{-2t(z+\delta)} L_0(z) + 2 e^{-t(z+\delta)} \left( \frac{1 - e^{-t(z+\delta)}}{\delta + z} \right) U_0(z) + \left( \frac{1 - e^{-t(\delta+z)}}{\delta + z} \right)^2 V(z) \right\}. \tag{5}$$

*Let* $H_t(x,y) = \frac{a_t^T}{\sqrt{N}} R(x) \frac{\Theta\Theta^T}{d} R(y) \frac{a_t}{\sqrt{N}}$, $Q_t(x,y) = \frac{a_t^T}{\sqrt{N}} R(x) \frac{\Theta\Theta^T}{d} R(y) \frac{Z^T Y}{N}$ *and* $W(x,y) = \frac{Y^T Z}{N} R(x) \frac{\Theta\Theta^T}{d} R(y) \frac{Z^T Y}{N}$. *For all* $t \leq 0$

$$h(t) = \mathcal{R}_{x,y} \left\{ e^{-t(2\delta+x+y)} \left( \frac{e^{t(\delta+y)}-1}{\delta+y} Q_0(x,y) + \frac{e^{t(\delta+x)}-1}{\delta+x} Q_0(y,x) \right) \right\}$$

$$+ \mathcal{R}_{x,y} \left\{ \frac{1-e^{-t(x+\delta)}}{x+\delta} \frac{1-e^{-t(y+\delta)}}{y+\delta} W(x,y) \right\} + \mathcal{R}_{x,y} \left\{ e^{-t(x+y+2\delta)} H_0(x,y) \right\}. \quad (6)$$

A similar but much simpler representation holds for the training error.

**Proposition 2.0.2 (Training error)** *With the same definitions than in proposition 2.0.1 we have*

$$\mathcal{H}_t^{train} = \frac{\|Y\|^2}{n} + \frac{1}{c} \mathcal{R}_z \left\{ (z+\delta)e^{-2t(z+\delta)} L_0(z) - 2e^{-2t(z+\delta)} U_0(z) - \frac{1-e^{-2t(\delta+z)}}{\delta+z} V(z) \right\}. \quad (7)$$

### 2.3 High-dimensional framework

The Cauchy integral representation involves a set of one-variable functions $\mathcal{S}_1 = \{G_0, K, L_0, U_0, V\} : \mathbb{C} \setminus \mathrm{Sp}(\frac{Z^T Z}{N}) \to \mathbb{C}$ and a set of two-variable functions $\mathcal{S}_2 = \{H_0, W, Q_0\} : \mathbb{C} \setminus \mathrm{Sp}(\frac{Z^T Z}{N}))^2 \to \mathbb{C}$ so that $g$, $h$, $l$ and thus also $\mathcal{H}^{\text{test}}$ and $\mathcal{H}^{\text{train}}$ are actually functions of $(t; \mathcal{S}_1, \mathcal{S}_2)$. Thus we can write for instance: $\mathcal{H}^{\text{test}}(a_t) = \mathcal{H}^{\text{test}}(t; \mathcal{S}_1, \mathcal{S}_2)$. We simplify the problem by considering the high-dimensional regime where $N, n, d \to \infty$ with ratios $\frac{N}{d} \to \psi$, $\frac{n}{d} \to \phi$ tending to fixed values of order one. In this regime we expect that the functions in $\mathcal{S}_1$ and $\mathcal{S}_2$ concentrate and can therefore be replaced by their averages over randomness. These averages can be carried out using recent progress in random matrix theory [30], [31], and we are able to compute pointwise asymptotic values of the functions in $\mathcal{S}_1$, $\mathcal{S}_2$, and eventually substitute them in the Cauchy integral representations for the training and test error. In general, rigorously showing concentration of the various functions involved is not easy and we will make the following assumptions:

**Assumptions 2.1** *In the high dimensional limit with $d, N, n \to \infty$ and $\frac{N}{d} \to \psi$, $\frac{n}{d} \to \phi$:*

1. *The random functions in $\mathcal{S}_1$, $\mathcal{S}_2$ are assumed to concentrate. We let $\bar{\mathcal{S}}_1 = \{\bar{G}_0, \bar{K}, \bar{L}_0, \bar{U}_0, \bar{V}\}$ and $\bar{\mathcal{S}}_2 = \{\bar{H}_0, \bar{W}, \bar{Q}_0\}$ be the pointwise limit of the functions.*

2. *There exists a bounded subset $\mathcal{C} \subset \mathbb{R}^+$ such that the functions in $\bar{\mathcal{S}}_1$ and $\bar{\mathcal{S}}_2$ are holomorphic on $\mathbb{C} \setminus \mathcal{C}$ and $(\mathbb{C} \setminus \mathcal{C})^2$ respectively*

3. *The gaussian equivalence principle (see sect. 4.2) can be applied to the limiting quantities.*

It is common that the closure of the spectrum of suitably normalized random matrices concentrates on a deterministic set. Thus the bounded set $\mathcal{C}$ can be understood as the limit of the finite interval $[0, \lim_d \max \mathrm{Sp}(\frac{Z^T Z}{N})]$. In the sequel we will distinguish the theoretical high-dimensional regime from the finite dimensional regime using the upper-bar notation.

**Definition 2.1 (High-dimensional framework)** *Under the assumptions 2.1, we define the theoretical test error $\bar{\mathcal{H}}_t^{test} = \mathcal{H}^{test}(t; \bar{\mathcal{S}}_1, \bar{\mathcal{S}}_2)$ and the theoretical training error $\bar{\mathcal{H}}_t^{train} = \mathcal{H}^{train}(t; \bar{\mathcal{S}}_1, \bar{\mathcal{S}}_2)$*

We conjecture that $\lim_d \mathcal{H}_t^{\text{train}} = \bar{\mathcal{H}}_t^{\text{train}}$ and $\lim_d \mathcal{H}_t^{\text{test}} = \bar{\mathcal{H}}_t^{\text{test}}$ at all times $t \in \mathbb{R}$. We verify that this conjecture stands experimentally for sufficiently large $d$ on different configurations (see additional figures in the SM). This also lends experimental support on the assumption 2.1. Furthermore we conjecture that the $d \to +\infty$ and $t \to \infty$ limits commute, namely $\lim_d \lim_t \mathcal{H}_t^{\text{train}} = \lim_t \bar{\mathcal{H}}_t^{\text{train}}$ and $\lim_d \lim_t \mathcal{H}_t^{\text{test}} = \lim_t \bar{\mathcal{H}}_t^{\text{test}}$.

## 3 Results and insights

### 3.1 Main results

In this section we provide the main results of this work: analytical formulas tracking the test and training errors during gradient flow of the random feature model for all times in the high-dimensional theoretical framework.

**Result 3.1** *Under the assumption 2.1, the theoretical test and training errors of definition 2.1 are given for all times $t \geq 0$ by the formulas*

$$\bar{\mathcal{H}}_t^{test} = 1 + s^2 - 2\mu\bar{g}(t) + \mu^2\bar{h}(t) + \nu^2\bar{l}(t), \tag{8}$$

$$\bar{\mathcal{H}}_t^{train} = 1 + s^2 + \frac{1}{c}\int_{\mathbb{R}}\left[(\delta+\omega)e^{-2t(\omega+\delta)}\rho_{\bar{L}_0}(\omega) - \frac{1-e^{-2t(\delta+\omega)}}{\delta+\omega}\rho_{\bar{V}}(\omega)\right]\mathrm{d}\omega, \tag{9}$$

*with $c = \frac{\phi}{\psi}$, $\delta = c\lambda$, and the functions $\bar{g}, \bar{h}, \bar{l}$ given by*

$$\bar{g}(t) = \int_{\mathbb{R}}\frac{1-e^{-t(\omega+\delta)}}{\omega+\delta}\rho_{\bar{K}}(\omega)\mathrm{d}\omega, \tag{10}$$

$$\bar{l}(t) = \int_{\mathbb{R}}\left[e^{-2t(\omega+\delta)}\rho_{\bar{L}_0}(\omega) + \left(\frac{1-e^{-t(\omega+\delta)}}{\omega+\delta}\right)^2\rho_{\bar{V}}(\omega)\right]\mathrm{d}\omega, \tag{11}$$

$$\bar{h}(t) = \iint_{\mathbb{R}^2}\left[e^{-t(u+v+2\delta)}\rho_{\bar{H}_0}(u,v) + \frac{1-e^{-t(u+\delta)}}{u+\delta}\frac{1-e^{-t(v+\delta)}}{v+\delta}\rho_{\bar{W}}(u,v)\right]\mathrm{d}u\mathrm{d}v, \tag{12}$$

*where the measures $\rho_{\bar{K}}, \rho_{\bar{L}_0}, \rho_{\bar{V}}, \rho_{\bar{H}_0}, \rho_{\bar{W}}$ (are possibly signed) are characterized by their Stieltjes transforms given by $\bar{K}, \bar{L}_0, \bar{V}, \bar{H}_0, \bar{W}$*

$$\begin{cases} \bar{K}(x) = t_1^x, \quad \bar{L}_0(x) = r^2g_1^x, \quad \bar{V}(x) = s^2\left(1+xg_1^x\right) + \left(c - h_4^x\right), \\ \bar{H}_0(x,y) = r^2q_1, \quad \bar{W}(x,y) = s^2cq_4 + q_2 \end{cases} \tag{13}$$

*where for each $x,y \in \mathbb{C}^+$ (the upper half complex plane) $g_1^x, h_4^x, t_1^x, g_1^y, h_4^y, t_1^y$ and $q_1, q_2, q_4, q_5$ (which depend symmetrically on $(x,y)$, e.g., $q_1 = q_1^{x,y} = q_1^{y,x}$) are solutions of a purely algebraic system of equations (see SM for the criterion to select the relevant solution)*

$$\begin{cases} 0 = \mu\psi g_1^x h_4^x - t_1^x \\ 0 = \mu\psi g_1^y h_4^y - t_1^y \\ 0 = (c - 1 - xg_1^x)\left(c - \mu^2\phi g_1^x h_4^x\right) - ch_4^x \\ 0 = (c - 1 - yg_1^y)\left(c - \mu^2\phi g_1^y h_4^y\right) - ch_4^y \\ 0 = 1 - g_1^x\left(\mu^2 h_4^x + (c-1-xg_1^x)\nu^2 - x\right) \\ 0 = 1 - g_1^y\left(\mu^2 h_4^y + (c-1-yg_1^y)\nu^2 - y\right) \\ 0 = -\mu^2 g_1^y q_2 + \mu^2 h_4^x q_1 + \mu g_1^y t_1^x + \mu g_1^y t_1^y - cg_1^y q_4\nu^2 - g_1^y - q_1 x + q_1\nu^2\left(c - g_1^x x - 1\right) \\ 0 = \mu\left(\phi - \psi g_1^x x - \psi\right)\left(-\mu g_1^x q_2 + \mu h_4^y q_1 + g_1^x t_1^y\right) + cq_4(1 - \mu t_1^y) - q_2 \\ 0 = -\mu^2\phi g_1^x(1 - \mu t_1^x)q_4 + \mu^2 q_5\left(c - g_1^y y - 1\right) - \nu^2\phi g_1^x q_4 - \phi q_4 + q_1\nu^2\left(\phi - \psi g_1^y y - \psi\right) \\ 0 = \psi(\mu^2\phi g_1^x g_1^y q_4 + \psi g_1^x g_1^y + q_1)(1 - \mu t_1^y) - \mu^2\psi g_1^x q_5\left(c - g_1^x x - 1\right) - q_5 \end{cases}$$

We can also deduce the limiting training error and test errors in the infinite time limit:

**Result 3.2** *In the limit $t \to \infty$ we find:*

$$\lim_{t\to\infty}\bar{\mathcal{H}}_t^{test} = 1 + s^2 - 2\mu\bar{K}(-\delta) + \mu^2\bar{W}(-\delta,-\delta) + \nu^2\frac{\mathrm{d}\bar{V}}{\mathrm{d}x}(-\delta), \quad \lim_{t\to\infty}\bar{\mathcal{H}}_t^{train} = 1 + s^2 - \frac{1}{c}\bar{V}(-\delta)$$

Interestingly, in the limit $t \to \infty$, the expressions become simpler and completely algebraic in the sense that we do not need to compute integrals (or double-integrals) over the supports of the eigenvalue distributions. It is not obvious to see on the analytical expressions that the result is the same as the algebraic expressions obtained in [17] but Fig. 2 shows an excellent match with simulation experiments. We note here that checking that two sets of complicated algebraic equations are equivalent is in general a non-trivial problem of computational algebraic geometry [35].

### 3.2 Insights and illustrations of results

The set of analytical formulas allows to compute numerically the measures $\rho_K, \rho_{L_0}, \rho_V, \rho_{H_0}, \rho_W$ and in turn the full time evolution of the test and training errors. The result matches the simulation of a large random feature model where $d$ is taken large as can be seen on Figs. 2 for the infinite time limit (experimental check of result 3.2) and additional figures in the SM (experimental check of result 3.1). Below we illustrate numerical computations obtained with analytical formulas of result 3.1

for various sets of parameters $(t, \mu, \nu, \psi, \phi, r, s, \lambda)$. For instance, we can freely choose two of these parameters and plot the generalization error in 3D as in Fig. 1, or as a heat-map in the following. We describe three important phenomena which are observed with our analysis.

*Double descent and early-stopping benefits:* while [17] mostly analyze the minimum least-squares estimator of the random feature model which displays the double-descent at $\psi = \phi$, we are predicting the whole time evolution of the gradient flow as in Fig. 3. We clearly observe the double-descent curve at $t = 10^{10}$ for $\psi = \phi$; but we now notice that if we stop the training earlier, say at times $1 < t < 10$, the generalization error performs better than the minimum least squares estimator. Actually, in the time interval $t \in (1, 10)$ for $\psi \approx \phi$ the test error even has a *dip or minimum* just before the spike develops. We also notice a two-steps descent structure with the test error which is non-existent in the training error and materializes long after the training error has stabilized in the overparameterized regime $\psi \gg \phi$. This is also reminiscent but not entirely similar to the abrupt *grokking* phenomenon described in [36].

*Triple descent:* We can observe a triple descent phenomenon materialized by two spikes as seen in Fig. 2 at $t = \infty$ (we also check that the theoretical result matches very well the empirical prediction of the minimum least squares estimator both for training and test errors). This triple descent phenomenon is already contained in the formulas of [17] (although not discussed in this reference) and has been analyzed in detail in [21]. The test error contains a so-called *linear spike* for $\phi = 1$ ($n = d$) and a *non-linear* spike for $\psi = \phi$ ($N = n$). The two spikes are often not seen together as this requires certain conditions to be met, and they tend to materialize together for specific values $\mu$, $\nu$ of the activation function where $\mu \gg \nu$. Here we further observe the evolution through time of the triple descent and the two spikes and how they develop in Fig. 4. There, we notice that the linear-spike seems to appear *earlier* than the non-linear one.

*Epoch-wise descent structures:* Important phenomena that we uncover here are two time-wise "descent structures". (i) As can be seen in Fig. 3, the test error develops a *double plateau* structure at widely different time scales in the over-parameterized regime ($\psi \gg \phi$) while there seems to be only one time scale for the training error. This kind of double plateau descent is different from the "usual" double-descent. (ii) Moreover, on Fig. 5 for well chosen parameters (in particular for noises with $s$ and $r$ "larger" and $\psi = 2\phi$), we can also observe an *elongated bump* (rather than a thin spike) for small $\lambda$'s. Notice the logarithmic time-scale which clearly shows that here we need to wait exponentially longer to attain the "second descent" after the bump. This is very reminiscent of the epoch-wise double descent described in [22] for deep networks (which happens on similar time scales).

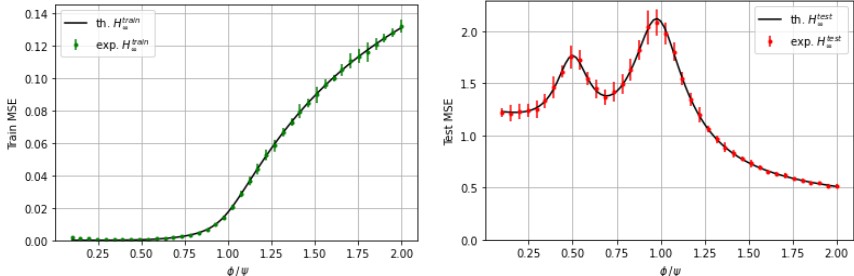

Figure 2: *Large time limit.* Analytical training error and test error profile with parameters $(\mu, \nu, \psi, r, s, \lambda) = (10, 1, 2, 1, 0.5, 0.01)$ compared with experimental least squares MSE with 40 data-points with $d = 5000$ (average of 10 instances with confidence bar at $2\sigma$)

## 4    Sketch of proofs and analytical derivations

The analysis is threefold. Firstly, we decompose the training and test errors in elementary terms and establish Cauchy's integral representation for each of them, as provided in proposition 2.0.1. A crucial advantage of this form is that it dissociates a scalar time-wise component and static matrix terms. Secondly, we switch to the high-dimensional framework where the matrix terms are substituted by their limit using the gaussian equivalence principle. Thirdly, we can compute the expectations of

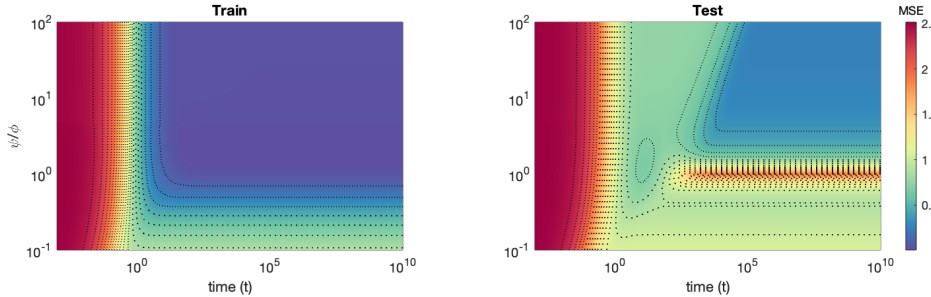

Figure 3: *Model-wise double descent.* Analytical training error and test error evolution with parameters $(\mu, \nu, \phi, r, s, \lambda) = (0.5, 0.3, 3, 2., 0.4, 0.001)$. Note that we vary the number of model parameters $(\psi)$.

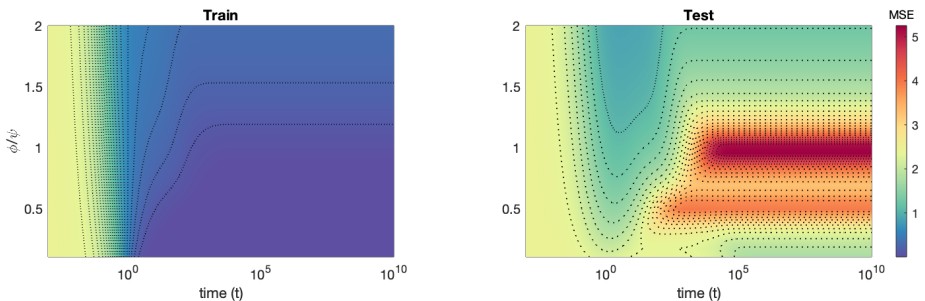

Figure 4: *Sample-wise descents.* Analytical training error and test error evolution with parameters $(\mu, \nu, \psi, r, s, \lambda) = (0.9, 0.1, 2, 1, 0.8, 0.0001)$. Note that we vary the number of samples $(\phi)$.

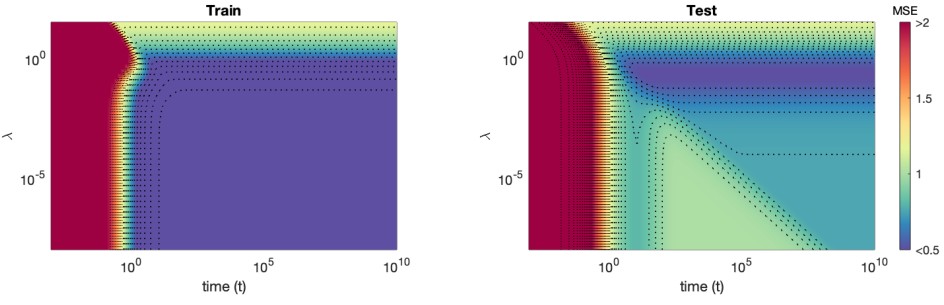

Figure 5: *Epoch-wise descent structures.* Analytical test error evolution with respect to different values of $\lambda$ $(\mu, \nu, \psi, \phi, r, s) = (0.5, 0.3, 6, 3, 2.0, 0.5)$. Here the ratio of number of parameters and samples is fixed.

matrix terms thanks to a random matrix technique based on linear pencils. In this section we only sketch the main ideas for each step and provide details in the supplementary material.

## 4.1 Cauchy's integral representation

We sketch the derivation for the test error and leave details to appendices. The derivation for the training error is entirely found in the SM. Expanding the square in Equ. (1) and carrying out averages we find Equ. (3) for $\mathcal{H}_t^{\text{test}}$ with $g(t) = \frac{\beta^T}{\sqrt{d}} \frac{\Theta^T}{\sqrt{d}} \frac{a_t}{\sqrt{N}}$, $h(t) = \|\frac{\Theta^T}{\sqrt{d}} \frac{a_t}{\sqrt{N}}\|^2$, and $l(t) = \|\frac{a_t}{\sqrt{N}}\|^2$ (see SM for this derivation).

We show how to derive the Cauchy integral representation for $g(t)$. For $h(t), l(t)$ the steps are similar and found in SM. Let us consider the function $(t, z) \mapsto G_t(z)$ as in 2.0.1. Then we have the relation $g(t) = \frac{-1}{2i\pi} \oint_\Gamma dz G_t(z)$ where $\Gamma$ is a loop in $\mathbb{C}$ enclosing the spectrum of $\frac{Z^T Z}{N}$. This can easily be seen by decomposing the symmetric $\frac{Z^T Z}{N}$ in an orthonormal basis $v_1, \ldots, v_N$ with the eigenvalues

$\lambda_1, \ldots, \lambda_N$: then we have $G_t(z) = \sum_{i=1}^{N} \frac{1}{\lambda_i - z} \left( \frac{\beta^T}{\sqrt{d}} \frac{\Theta^T}{\sqrt{d}} v_i v_i^T \frac{a_t}{\sqrt{N}} \right)$ and because $\lambda_i$ are all encircled by $\Gamma$, we find $-\oint_\Gamma \frac{dz}{2\pi i} G_t(z) = \sum_{i=1}^{N} \frac{\beta^T}{\sqrt{d}} \frac{\Theta^T}{\sqrt{d}} v_i v_i^T \frac{a_t}{\sqrt{N}} = g(t)$. Now, the ODE derived for $a_t$ in (2), can be written slightly differently using the fact that $\frac{Z^T Z}{N} = R(z)^{-1} + zI$ for any $z$ outside $\mathrm{Sp}(\frac{Z^T Z}{N})$. Namely, $\frac{da_t}{dt} = \frac{Z^T Y}{\sqrt{N}} - R(z)^{-1} a_t - (z + \delta) a_t$. Then, we can derive an integro-differential equation for $G_t(z)$ involving $g(t)$ and $K(z)$:

$$\frac{\partial_t G_t(z)}{\partial t} = K(z) - g(t) - (z + \delta) G_t(z) \tag{14}$$

In the following, we let $\mathcal{L}$ be the Laplace transform operator $(\mathcal{L}f)(p) = \int_0^{+\infty} dt\, e^{-pt} f(t)$, $\mathrm{Re}\, p$ large enough. Note that the contour integral is performed over a compact set $\Gamma$ so for $\mathrm{Re}\, p$ large enough, by Fubini's theorem, the operations $\mathcal{L}$ and $\mathcal{R}_z$ commute. Applying $\mathcal{L}$ to (14) and rearranging terms we find for $\mathrm{Re}(p + z + \delta) \neq 0$:

$$\mathcal{L}G_p(z) = \frac{G_0(z)}{p + z + \delta} + \frac{K(z)}{p(p + z + \delta)} - \frac{\mathcal{L}g(p)}{p + z + \delta} \tag{15}$$

Now, we can always choose $\Gamma$ such that $-(p+\delta)$ is outside of the contour if we assume $\mathrm{Re}(p+\delta) > 0$ (since $\min_i \lambda_i \geq 0$). Thus, applying $\mathcal{R}_z$ to (15) nullifies the last term because the pole is outside $\Gamma$, and using commutativity $\mathcal{R}_z \mathcal{L}G_p = \mathcal{L}\mathcal{R}_z G_p$,

$$\mathcal{R}_z \mathcal{L}G_p = \mathcal{L}\mathcal{R}_z \left\{ e^{-t(z+\delta)} G_0(z) + \frac{1 - e^{-t(z+\delta)}}{z + \delta} K(z) \right\} = \mathcal{L}g(p). \tag{16}$$

Finally, using the inverse Laplace transform leads to (4).

## 4.2 Gaussian equivalence principle

The matrix terms must be estimated in the limit $d \to \infty$ with $\{\beta, a_0, \xi, \Theta, X\}$ all independently distributed. As per assumptions 2.1 all the matrix terms in $\mathcal{S}_1, \mathcal{S}_2$ are assumed to concentrate. So for instance we assume that the following limit exists $\bar{K}(z) \equiv \lim_{d \to \infty} K(z) = \lim_{d \to \infty} \mathbb{E}_{\beta, \xi, \Theta, X}[K(z)]$. Using cyclicity of the trace we easily perform averages over $\beta, \xi$ to find

$$\bar{K}(z) = \lim_d \mathbb{E}_{\beta, \Theta, X} \mathrm{Tr} \left[ \frac{\Theta^T}{\sqrt{d}} R(z) \frac{Z^T X}{N} \frac{\beta \beta^T}{d} \right] = \lim_d \frac{1}{d} \mathbb{E}_{\beta, \Theta, X} \mathrm{Tr} \left[ \frac{\Theta^T}{\sqrt{d}} R(z) \frac{Z^T X}{N} \right]. \tag{17}$$

After these reductions, the expressions of all functions in $\bar{\mathcal{S}}_1, \bar{\mathcal{S}}_2$ essentially involve products of random matrices $\Theta$, $X$ and pointwise applications of the non-linear activation $\sigma$. This can be further reduced to simpler algebraic expressions using the *gaussian equivalence principle*. This principle states that: *there exists a standard gaussian random matrix $\Omega \in \mathbb{R}^{n \times N}$ independent of $\{X, \Theta\}$ such that in the infinite dimensional limit we can make the substitution $Z = \sigma\left(d^{-1/2} X \Theta^T\right) \longrightarrow \mu d^{-1/2} X \Theta^T + \nu \Omega$ in the expressions of all functions in $\bar{\mathcal{S}}_1, \bar{\mathcal{S}}_2$*. This approach is quite general and is well described in [29, 37] (and formerly in [33] and [34]). Thus it remains to compute expectations of traces containing only products, and inverses of products and sums, of gaussian matrices.

## 4.3 Expectations over random matrices using linear pencils

We explain how to compute the limit of (17) once the gaussian equivalence principle has been applied. A powerful approach is to design a so-called *linear pencil*. In the present context this is a suitable block-matrix containing gaussian random matrices and multiples of the identity matrix, for which full block-inversion gives back the products of terms in the traces that are being sought. This approach has been described in [30, 31, 38]. We have found a suitable linear pencil which contains fortuitously *all* the terms required in $\bar{\mathcal{S}}_1, \bar{\mathcal{S}}_2$. It is described by the $13 \times 13$ *block-matrix $M$*, and pursuing with

our example, we get for instance with the block $(7, 12)$ that $\lim_d K(y) = \lim_d \frac{1}{d}\mathrm{Tr}[(M^{-1})^{(7,12)}]$

$$M = \begin{bmatrix}
-xI & -\mu\frac{\Theta}{\sqrt{d}} & -I & 0 & 0 & 0 & \frac{\Theta}{\sqrt{d}} & 0 & 0 & 0 & 0 & 0 & 0 \\
0 & I & 0 & \frac{X^T}{\sqrt{N}} & 0 & 0 & 0 & 0 & 0 & 0 & 0 & 0 & 0 \\
0 & 0 & I & \nu\frac{\Omega^T}{\sqrt{N}} & 0 & 0 & 0 & 0 & 0 & 0 & 0 & 0 & 0 \\
0 & 0 & 0 & I & \frac{X}{\sqrt{N}} & \nu\frac{\Omega}{\sqrt{N}} & 0 & 0 & 0 & 0 & 0 & 0 & 0 \\
\mu\frac{\Theta^T}{\sqrt{d}} & 0 & 0 & 0 & I & 0 & 0 & 0 & 0 & 0 & 0 & 0 & 0 \\
I & 0 & 0 & 0 & 0 & I & 0 & 0 & 0 & 0 & 0 & 0 & 0 \\
0 & 0 & 0 & 0 & 0 & 0 & I & 0 & 0 & 0 & 0 & 0 & \frac{\Theta^T}{\sqrt{d}} \\
0 & 0 & 0 & 0 & 0 & 0 & 0 & I & 0 & \nu\frac{\Omega^T}{\sqrt{N}} & 0 & 0 & 0 \\
0 & 0 & 0 & 0 & 0 & 0 & 0 & 0 & I & \frac{X^T}{\sqrt{N}} & 0 & 0 & 0 \\
0 & 0 & 0 & 0 & 0 & 0 & 0 & 0 & 0 & I & \nu\frac{\Omega}{\sqrt{N}} & \frac{X}{\sqrt{N}} & 0 \\
0 & 0 & 0 & 0 & 0 & 0 & 0 & 0 & 0 & 0 & I & 0 & -I \\
0 & 0 & 0 & 0 & 0 & 0 & 0 & 0 & 0 & 0 & 0 & I & -\mu\frac{\Theta^T}{\sqrt{d}} \\
0 & 0 & 0 & 0 & 0 & 0 & 0 & I & \mu\frac{\Theta}{\sqrt{d}} & 0 & 0 & 0 & -yI
\end{bmatrix}$$

Next, the great advantage of the linear pencil is that (as described in [30, 31, 38]) it allows to write a fixed point equation $F(G) = G$ for a "small" $13 \times 13$ matrix $G$ with *scalar* matrix elements. We also provide in the SM an independent derivation of the fixed point equations using the replica method (a technique from statistical physics [39]). The components of $G$ are linked to the limiting traces of the blocks of $M^{-1}$ as in $[G]_{2,7} = \bar{K}(z)$. The action of $F$ can be completely described as an algebraic function leading to (a priori) $13 \times 13 = 169$ equations over the matrix elements of $G$. The number of equations can be immediately reduced to 39 because many elements vanish, and with the help of a computer algebra system the number of equations can be further brought down to 10. We refer to the SM for all the details about the method.

## 5 Conclusion

We believe that our analysis could be extended to study the learning of non-linear functions, the effect of multilayered structures, and potentially different layers such as convolutions, as long as they are not learned. A challenging task is to extend the present methods to learned multilayers. A further question is the application of our analysis in teacher-student scenarios with realistic datasets (See [40, 37]).

Finally we wish to point out that a comparison of the approach of the present paper (and the similar but simpler one of [27]) with the dynamical mean-field theory (DMFT) approach of statistical physics remains to be investigated. DMFT has a long history originating in studies of complex systems (turbulent fluids, spin glasses) where one eventually derives a set of complicated integro-differential equations for suitable correlation and response functions capturing the whole dynamics of the system (we refer to the recent book [41] and references therein). This is a powerful formalism but the integral equations must usually be solved entirely numerically which itself is not a trivial task. For problems close to the present context (neural networks, generalized linear models, phase retrieval) DMFT has been developed in the recent works [42, 43, 44, 45, 46]. We think that comprehensively comparing this formalism with the present approach is an interesting open problem. It would be desirable to connect the DMFT equations to our closed form solutions for the training and generalization errors expressed in terms of a set of algebraic equations of suitable Stieltjes transforms.

## Acknowledgments and Disclosure of Funding

The work of A. B has been supported by Swiss National Science Foundation grant no 200020 182517.

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
