# A Test Error substitutions

The test error $\mathcal{H}_t^{\text{test}}$ in (1) can be expanded into smaller terms

$$\mathcal{H}_t^{\text{test}} = \mathbb{E}_{x_0}[y(x_0)^2] - 2\mathbb{E}_{x_0}[y(x_0)\hat{y}_t(x_0)] + \mathbb{E}_{x_0}[\hat{y}_t(x_0)^2]$$

$$= \mathbb{E}_{x_0}[y(x_0)^2] - 2\frac{\beta^T}{\sqrt{d}}\mathbb{E}_{x_0}[x_0 z(x_0)^T]\frac{a_t}{\sqrt{N}} + \frac{a_t^T}{\sqrt{N}}\mathbb{E}_{x_0}[z(x_0)z(x_0)^T]\frac{a_t}{\sqrt{N}}. \tag{18}$$

The random noise $\epsilon$ from $y(x_0)$ only impacts the first term on the right hand side with $\mathbb{E}_{x_0}[y(x_0)^2] = 1 + s^2$. Using further $q(t) = \frac{\beta^T}{\sqrt{d}}\mathbb{E}_{x_0}[x_0 z(x_0)^T]\frac{a_t}{\sqrt{N}}$ and $p(t) = \frac{a_t^T}{\sqrt{N}}\mathbb{E}_{x_0}[z(x_0)z(x_0)^T]\frac{a_t}{\sqrt{N}}$, we write $\mathcal{H}_t^{\text{test}} = 1 + s^2 - 2q(t) + p(t)$.

We provide analytical arguments to justify the formula (3) showing that:

$$q(t) = \mu g(t) + o_d(1) \tag{19}$$
$$p(t) = \mu^2 h(t) + \nu^2 l(t) + o_d(1) \tag{20}$$

with

$$g(t) = \frac{\beta^T}{\sqrt{d}}\frac{\Theta^T}{\sqrt{d}}\frac{a_t}{\sqrt{N}}, \quad l(t) = \left\|\frac{a_t}{\sqrt{N}}\right\|^2, \quad h(t) = \left\|\frac{\Theta^T}{\sqrt{d}}\frac{a_t}{\sqrt{N}}\right\|^2 \tag{21}$$

and where $\lim_{d\to+\infty} o_d(1) = 0$ with probability tending to one when $d \to +\infty$. The arguments below are based further on the prior assumption that the $(\theta_i/\sqrt{d})$ are sampled uniformly on the hyper-sphere of radius 1. We will assume further that these results can be extended in our setting with $\theta_i$ sampled from a gaussian distribution. Notice that this is a reasonable assumption because $\|\theta_i\|^2/d$ is a $\chi^2$ distribution of mean 1 and variance $\frac{2}{d}$.

## A.1 limit of $q(t)$

We decompose our activation function as $\sigma(x) = \mu x + \nu \sigma^\perp(x)$ where $\sigma^\perp \in \text{Span}(H_{e_i})_{i\geq 2}$. In other words, we have $\mathbb{E}_G[\sigma^\perp(G)] = \mathbb{E}_G[\sigma^\perp(G)G] = 0$ and $\mathbb{E}_G[\sigma^\perp(G)^2] = 1$. Notice that conditional on $(\theta_i)_i$ sampled on the sphere of radius $\sqrt{d}$, we have for all $i \in \{1, \dots, N\}$ that $u_i \equiv \frac{\theta_i^T x_0}{\sqrt{d}} \underset{x_0}{\sim} \mathcal{N}(0,1)$, and for all $j \in \{1, \dots, N\}$, we have $\text{Cov}(u_i, u_j) = \frac{\theta_i^T \theta_j}{d} = \left[\frac{\Theta\Theta^T}{d}\right]_{i,j}$. Similarly, for any $l \in \{1, \dots, d\}$ we have $\text{Cov}(u_j, [x_0]_l) = \frac{[\theta_j]_l}{\sqrt{d}}$.. Now, using the Mehler-Kernel formula, we have

$$\mathbb{E}_{x_0}\left[[x_0]_l[z(x_0)]_j\right] = \sum_{k\geq 0}\frac{1}{k!}\left(\text{Cov}(u_j, [x_0]_l)\right)^k \mathbb{E}_{x_0}\left[x_0 H_{e_k}(x_0)\right]\mathbb{E}_{u_j}\left[\sigma(u_j)H_{e_k}(u_j)\right] \tag{22}$$

which does not vanish only for $k = 1$ due to the first expectation on the RHS. Thus

$$\mathbb{E}_{x_0}\left[[x_0]_l[z(x_0)]_j\right] = \frac{[\theta_j]_l}{\sqrt{d}}\mu \tag{23}$$

and hence we find that $q(t) = \frac{\beta^T}{\sqrt{d}}\mathbb{E}_{x_0}\left[x_0 z(x_0)^T\right]\frac{a_t}{\sqrt{N}} = \mu\frac{\beta^T}{\sqrt{d}}\frac{\Theta^T}{\sqrt{d}}\frac{a_t}{\sqrt{N}}$.

The result ought not be exact anymore when $(\theta_i)$ are sampled from a normal distribution, and we make the assumption that we can account for a correction term $o_d(1)$ which goes to 0 as $d$ grows to infinity, hence $q(t) = \mu g(t) + o_d(1)$ in general.

## A.2 limit of $p(t)$

Similarly for $p(t)$, we evaluate the kernel $U_{i,j} = \mathbb{E}_{x_0}\left[[z(x_0)]_i[z(x_0)]_j\right]$ for which the Mehler-Kernel formula provides

$$\begin{aligned}U_{i,j} &= \sum_{k\geq 0}\frac{1}{k!}\left(\text{Cov}(u_i, u_j)\right)^k \mathbb{E}_{u_i}\left[\sigma(u_i)H_{e_k}(u_i)\right]^2 \\ &= \mu^2\text{Cov}(u_i, u_j) + \nu^2\sum_{k\geq 2}\frac{(\text{Cov}(u_i, u_j))^k}{k!}\mathbb{E}_{u_i}\left[\sigma^\perp(u_i)H_{e_k}(u_i)\right]^2.\end{aligned} \tag{24}$$

Intuitively, the terms $(\mathrm{Cov}(u_i, u_j))^k$ for $k \geq 2$ are on a smaller order in $d$ compared to $\mathrm{Cov}(u_i, u_j)$ when $i \neq j$. We refer the reader to Lemma C.7 in [17] where it is shown with some additional assumptions on $\sigma$ (weakly differentiable with $\exists c_0, c_1, \forall x > 0, |\sigma(x)|, |\sigma'(x)| \leq c_0 e^{c_1 x}$) that:

$$\mathbb{E}_\Theta \left[ \left\| U - \mu^2 \frac{\Theta\Theta^T}{d} - \nu^2 I_N \right\|_{\mathrm{op}} \right] = o_d(1). \tag{25}$$

Therefore, we can bound:

$$
\begin{aligned}
|p(t) - \mu^2 h(t) - \nu^2 l(t)| &= \left| \left\langle \frac{a_t^T}{\sqrt{N}}, \left( U - \mu^2 \frac{\Theta\Theta^T}{d} - \nu^2 \right) \frac{a_t^T}{\sqrt{N}} \right\rangle \right| \\
&\leq \left\| \frac{a_t}{\sqrt{N}} \right\| \cdot \left\| U - \mu^2 \frac{\Theta\Theta^T}{d} - \nu^2 I_N \right\|_{\mathrm{op}} \cdot \left\| \frac{a_t}{\sqrt{N}} \right\| \\
&= l(t) \left\| U - \mu^2 \frac{\Theta\Theta^T}{d} - \nu^2 I_N \right\|_{\mathrm{op}}.
\end{aligned} \tag{26}
$$

As per the general assumptions 2.1, $l(t)$ concentrates to a finite quantity $\bar{l}(t)$ at all times as $d$ grows to infinity (that $\bar{l}(t)$ is finite is explicitly checked by the anlytical computations of the generalization error). Thus by Markov's inequality we have at any fixed time $t$, $|p(t) - \mu^2 h(t) - \nu^2 l(t)| = o_d(1)$ with probability tending to one as $d \to +\infty$.

Notice also that we assume as before that $o_d(1)$ also contains the correction added when $(\theta_i)$ are sampled from a normal distribution.

## B  Cauchy's integral representation formula

In this section we complete the proof of propositions 2.0.1 and 2.0.2. We show how to derive the Cauchy integral representation of the two functions $l(t)$ and $h(t)$ by similar analysis of Sect. 4.1 for the representation of $g(t)$.

### B.1  Representation formula for $l(t)$

We define the function $L_t(z) = \frac{a_t^T}{\sqrt{N}} R(z) \frac{a_t}{\sqrt{N}}$ and the auxiliary functions $U_t(z) = \frac{Y^T Z}{N} R(z) \frac{a_t}{\sqrt{N}}$ and $V(z) = \frac{Y^T Z}{N} R(z) \frac{Z^T Y}{N}$. We find a set of 2 integro-differential equations using the gradient flow equation for $\frac{da_t}{dt}$ (as in the derivation of 14)

$$
\begin{aligned}
\frac{1}{2} \frac{\partial L_t(z)}{\partial t} &= U_t(z) - l(t) - (z + \delta) L_t(z) \\
\frac{\partial_t U_t(z)}{\partial t} &= V(z) - \mathcal{R}_z U_t - (z + \delta) U_t(z)
\end{aligned} \tag{27}
$$

Similarly $G_t(z)$ and $g(t)$, we also have that $l(t) = -\oint_\Gamma \frac{dz}{2i\pi} L_t(z) = \mathcal{R}_z L_t$. So we get a pair of integro-differential equations in this case (wheras for $G_t(z)$ we had only one such equation). However, we have one additional differential equation in this case. Pursuing with the Laplace transform operator[1] the equations (27) become

$$
\begin{aligned}
\mathcal{L}L_p(z) &= \frac{1}{\frac{1}{2}p + z + \delta} \left( \frac{1}{2} L_0(z) + \mathcal{L}U_p(z) - \mathcal{L}l(p) \right) \\
\mathcal{L}U_p(z) &= \frac{1}{p + z + \delta} \left( U_0(z) + \frac{V(z)}{p} - \mathcal{L}\mathcal{R}_z U_p \right)
\end{aligned} \tag{28}
$$

and re-injecting $\mathcal{L}U_p$ from the second equation into the first equation we find

$$\mathcal{L}L_p(z) = \frac{1}{\frac{1}{2}p + z + \delta} \left( \frac{L_0(z)}{2} - \mathcal{L}l(p) \right) + \frac{1}{(\frac{1}{2}p + z + \delta)(p + z + \delta)} \left( U_0(z) + \frac{V(z)}{p} - \mathcal{L}\mathcal{R}_z U_p \right). \tag{29}$$

---

[1] Defined as $(\mathcal{L}f)(p) = \int_0^{+\infty} dt\, e^{-pt} f(t)$ for $\mathrm{Re}\, p$ large enough. We also use the notation $\mathcal{L}f_p$ to mean $(\mathcal{L}f)(p)$ specially when there are other variables involved. For example $\mathcal{L}L_p(z) = \int_0^{+\infty} dt\, e^{-pt} L_t(z)$.

With similar considerations as before, with $p$ large enough to have $-\delta$ is outside the loop $\Gamma$, we see the terms $\mathcal{L}l(p)$ and $\mathcal{L}\mathcal{R}_z U_p$ don't contribute to the former equation when the operator $\mathcal{R}_z$ is applied

$$\mathcal{R}_z \mathcal{L} L_p(z) = \mathcal{R}_z \left\{ \frac{1}{2} \frac{L_0(z)}{\frac{1}{2}p + z + \delta} + \frac{1}{(\frac{1}{2}p + z + \delta)(p + z + \delta)} \left( U_0(z) + \frac{V(z)}{p} \right) \right\}. \tag{30}$$

Finally, there remains to use the commutativity of $\mathcal{R}_z$ and $\mathcal{L}$ (for $\operatorname{Re} p$ large enough by Fubini's theorem) and compute the inverse Laplace transforms to find

$$l(t) = \mathcal{R}_z \left\{ e^{-2t(z+\delta)} \left[ L_0(z) + 2 \frac{e^{t(\delta+z)} - 1}{\delta + z} U_0(z) + \left( \frac{e^{t(\delta+z)} - 1}{\delta + z} \right)^2 V(z) \right] \right\} \tag{31}$$

Expanding further the terms individually

$$l(t) = \mathcal{R}_z \left\{ e^{-2t(z+\delta)} L_0(z) + 2e^{-t(z+\delta)} \left( \frac{1 - e^{-t(z+\delta)}}{\delta + z} \right) U_0(z) + \left( \frac{1 - e^{-t(\delta+z)}}{\delta + z} \right)^2 V(z) \right\}. \tag{32}$$

We end-up (as for $g(t)$) with an expression where the time dependence is decoupled from random matrix expressions.

## B.2 Representation formula for $h(t)$

The last term requires additional considerations. We will now use a double contour $\Gamma_x, \Gamma_y$ enclosing the eigenvalues of $\frac{Z^T Z}{\sqrt{N}}$ and such that $\Gamma_x \cap \Gamma_y = \emptyset$. We consider the operators $\mathcal{R}_x, \mathcal{R}_y$ associated to each contour. Contrary to the previous two representations, when computing the multiple derivatives $h^{(k)}(t)$, due to the $\Theta$ matrix in $h(t)$, there appears pairs of matrices $\frac{Z^T Z}{\sqrt{N}}$. In terms of generating functions, this translates into a "2-variable resolvent" functions

$$H_t(x, y) = \frac{a_t^T}{\sqrt{N}} R(x) \frac{\Theta \Theta^T}{d} R(y) \frac{a_t}{\sqrt{N}}, \tag{33}$$

which has the property $h(t) = \mathcal{R}_{x,y} H_t$, and two auxiliary functions

$$Q_t(x, y) = \frac{a_t^T}{\sqrt{N}} R(x) \frac{\Theta \Theta^T}{d} R(y) \frac{Z^T Y}{N}, \quad \text{and} \quad W(x, y) = \frac{Y^T Z}{N} R(x) \frac{\Theta \Theta^T}{d} R(y) \frac{Z^T Y}{N}. \tag{34}$$

Using the former method for equation (27) leads to the following integro-differential equations:

$$\frac{\partial H_t(x, y)}{\partial t} = Q_t(x, y) + Q_t(y, x) - \mathcal{R}_x H_t(y) - \mathcal{R}_y H_t(x) - (x + y + 2\delta) H_t(x, y)$$
$$\frac{\partial Q_t(x, y)}{\partial t} = W(x, y) - \mathcal{R}_x Q_t(y) - (x + \delta) Q_t(x, y) \tag{35}$$

Then the Laplace transform on the first equation reads

$$\mathcal{L} H_p(x, y) = \frac{1}{p + x + y + 2\delta} \left[ H_0(x, y) + \mathcal{L} \left\{ Q_t(x, y) + Q_t(y, x) - \mathcal{R}_x H_t(y) - \mathcal{R}_y H_t(x) \right\} \right]. \tag{36}$$

Notice that $\mathcal{R}_x$ and $\mathcal{R}_y$ commute with each other as being integrals over a compact set $\Gamma_x, \Gamma_y$ respectively. So by Fubini we can name indifferently $\mathcal{R}_{x,y} = \mathcal{R}_x \mathcal{R}_y = \mathcal{R}_y \mathcal{R}_x$. Notice also that $\mathcal{R}_x H_t(y)$ is not a function of $x$ anymore, thus for $p$ large enough to have $|2\delta + x + y| > 0$ for all $(x, y) \in \Gamma_x \times \Gamma_y$, we find

$$\mathcal{R}_{x,y} \left\{ \frac{\mathcal{R}_x H_t(y)}{p + x + y + 2\delta} \right\} = \mathcal{R}_y \left\{ \mathcal{R}_x \left\{ \frac{\mathcal{R}_x H_t(y)}{p + x + y + 2\delta} \right\} \right\} = \mathcal{R}_y \{0\} = 0. \tag{37}$$

Symmetrically, the same statement can be made for $\mathcal{R}_y H_t(x)$, so applying the operator $\mathcal{R}_{x,y}$ and the result (37) to (36) we find

$$\mathcal{R}_{x,y} \mathcal{L} H_p(x, y) = \mathcal{R}_{x,y} \left\{ \frac{H_0(x, y) + \mathcal{L} Q_p(x, y) + \mathcal{L} Q_p(y, x)}{p + x + y + 2\delta} \right\}. \tag{38}$$

Finally, we have $\mathcal{R}_{x,y}\mathcal{L}H_p(x,y) = \mathcal{L}\mathcal{R}_{x,y}H_p(x,y) = \mathcal{L}h(p)$. The Laplace transform of the second equation of (35) provides

$$\mathcal{L}Q_p(x,y) = \frac{1}{p+x+\delta}\left(Q_0(x,y) + \frac{W(x,y)}{p} - \mathcal{R}_x\mathcal{L}Q_p(y)\right).\tag{39}$$

Before injecting this equation into (38) (and its symmetrical result in $x$ and $y$), notice that one term will not contribute under the operator $\mathcal{R}_{x,y}$

$$\mathcal{R}_{x,y}\left\{\frac{\mathcal{R}_x\mathcal{L}Q_p(y)}{(p+x+y+2\delta)(p+x+\delta)}\right\} = \mathcal{R}_y\{0\} = 0\tag{40}$$

and finally, using $W(x,y) = W(y,x)$, we obtain

$$\mathcal{L}h(p) = \mathcal{R}_{x,y}\left\{\frac{1}{p+x+y+2\delta}\left(H_0(x,y) + \frac{Q_0(x,y)+\frac{W(x,y)}{p}}{p+x+\delta} + \frac{Q_0(y,x)+\frac{W(x,y)}{p}}{p+y+\delta}\right)\right\}.\tag{41}$$

Eventually, applying inverse Laplace transform we get the representation

$$\begin{aligned}h(t) &= \mathcal{R}_{x,y}\left\{e^{-t(x+y+2\delta)}H_0(x,y)\right\}\\&+ \mathcal{R}_{x,y}\left\{e^{-t(2\delta+x+y)}\left(\frac{e^{t(\delta+y)}-1}{\delta+y}Q_0(x,y) + \frac{e^{t(\delta+x)}-1}{\delta+x}Q_0(y,x)\right)\right\}\\&+ \mathcal{R}_{x,y}\left\{\frac{1-e^{-t(x+\delta)}}{x+\delta}\frac{1-e^{-t(y+\delta)}}{y+\delta}W(x,y)\right\}\end{aligned}\tag{42}$$

### B.3 Remark on the consistency with the minimum least squares estimator

It can be seen, at least formally, that the integral representation formula correctly retrieves the minimum least-squares estimator formulas in the limit $t \to \infty$. Indeed, commuting $\lim_t$ and $\mathcal{R}_z$ we find

$$\begin{aligned}\lim_{t\to+\infty}g(t) &= \mathcal{R}_z\left\{\frac{1}{z+\delta}K(z)\right\} = \sum_{i=1}^N\frac{\beta^T}{\sqrt{d}}v_i\mathcal{R}_z\left\{\frac{1}{(\lambda_i+z)(z+\delta)}\right\}v_i^T\frac{Z^TY}{N}\\&= \sum_{i=1}^N\frac{\beta^T}{\sqrt{d}}\frac{v_iv_i^T}{(\lambda_i-\delta)}\frac{Z^TY}{N} = K(-\delta).\end{aligned}\tag{43}$$

On the other hand, we expect

$$\lim_{t\to+\infty}g(t) = \lim_t\frac{\beta^T}{\sqrt{d}}\frac{\Theta^T}{\sqrt{d}}a_t = \frac{\beta^T}{\sqrt{d}}\frac{\Theta^T}{\sqrt{d}}a_\infty\tag{44}$$

with $a_\infty$ defined as the minimum least-squares estimator. Thus, we clearly have:

$$\frac{\beta^T}{\sqrt{d}}\frac{\Theta^T}{\sqrt{d}}a_\infty = \frac{\beta^T}{\sqrt{d}}\frac{\Theta^T}{\sqrt{d}}\left(\frac{Z^TZ}{N}+\delta I\right)^{-1}\frac{Z^T}{\sqrt{N}}\frac{Y}{\sqrt{N}} = K(-\delta)\tag{45}$$

The same calculations can be done on each term $h(t), l(t)$.

### B.4 Representation formula for the training error

The derivation of $\mathcal{H}_t^{\text{train}}$ is quite straightforward based on the previous terms derived for the test error. Firstly, expanding the expression of $\mathcal{H}_t^{\text{train}}$ we get:

$$\mathcal{H}_t^{\text{train}} = \frac{1}{n}\left\|Y - Z\frac{a_t}{\sqrt{N}}\right\|^2 + \lambda\left\|\frac{a_t}{\sqrt{N}}\right\|^2 = \frac{\|Y\|^2}{n} - \frac{2}{n}Y^T\frac{Za_t}{\sqrt{N}} + \frac{1}{n}\left\|\frac{Za_t}{\sqrt{N}}\right\|^2 + \frac{\delta}{c}\left\|\frac{a_t}{\sqrt{N}}\right\|^2\tag{46}$$

Reusing the function $U_t(z)$ from Sect. B.1, and defining $u(t) = \mathcal{R}_zU_t(z) = \frac{1}{N}Y^T\frac{Za_t}{\sqrt{N}}$ and $\tilde{h}(t) = \frac{1}{N}\left\|\frac{Za_t}{\sqrt{N}}\right\|^2$, we get:

$$\mathcal{H}_t^{\text{train}} = \frac{\|Y\|^2}{n} + \frac{1}{c}\left(-2u(t) + \tilde{h}(t) + \delta l(t)\right)\tag{47}$$

Furthermore, reusing the differential equation found for $U_t(z)$, a simpler solution can be extracted for $u(t)$:

$$u(t) = \mathcal{R}_z \left\{ e^{-t(z+\delta)} U_0(z) + \frac{1 - e^{-t(z+\delta)}}{z + \delta} V(z) \right\} \tag{48}$$

The second term $\tilde{h}(t)$ can also be derived from the expression $L_t(z)$ which is also defined in appendix B.1. We find $\tilde{h}(t) = \mathcal{R}_z\{ zL_t(z) \}$. Hence the terms $\delta l(t)$ and $\tilde{h}(t)$ can be grouped together with $\tilde{h}(t) + \delta l(t) = \mathcal{R}_z\{ (z + \delta)L_t(z) \}$. Expanding from the expression of $\mathcal{R}_z \mathcal{L} L_t(z)$ we find

$$(\tilde{h} + \delta l)(t) = \mathcal{R}_z \Big\{ (z + \delta)e^{-2t(z+\delta)} L_0(z) + 2e^{-t(z+\delta)} \left( 1 - e^{-t(z+\delta)} \right) U_0(z)$$

$$+ \frac{\left( 1 - e^{-t(\delta+z)} \right)^2}{\delta + z} V(z) \Big\}. \tag{49}$$

Remarkably, all the terms can be summed together in (47) and we retrieve a simpler expression

$$\mathcal{H}_t^{\text{train}} = \frac{\|Y\|^2}{n} + \frac{1}{c} \mathcal{R}_z \left\{ (z + \delta)e^{-2t(z+\delta)} L_0(z) - 2e^{-2t(z+\delta)} U_0(z) - \frac{1 - e^{-2t(\delta+z)}}{\delta + z} V(z) \right\}. \tag{50}$$

## C High-dimensional limit

In this appendix we use assumption 2.1 in section 2.3 to compute limiting expressions of traces.

As $d \to \infty$, the mean of $a_0$ or $\beta$ converges two 0. Let's consider the auxiliary functions $U_0(z), G_0(z), Q_0(x,y)$. These three terms have only occurrence of $a_0$ and $\beta$ on each side of the matrix-vector multiplication composition (notice $\beta$ is also included in the term $Y$): they can be written in the form $F(H) = \frac{a_0^T}{\sqrt{N}} H \frac{\beta}{\sqrt{d}}$ where $H$ is a random matrix independent of $a_0, \beta$. For instance we have $G_0(z) = F\left( R(z) \frac{\Theta}{\sqrt{d}} \right)$. As the mean of $F(H)$ is precisely 0, assuming concentration, we have that these terms go to 0 when $d \to \infty$. The same considerations can be applied to the term $\xi$ from $Y$.

Besides, when a vector such as $a_0$ is expressed on both side of another expression such as $F(H) = \frac{a_0^T}{\sqrt{N}} H \frac{a_0}{\sqrt{N}}$, it can still be rewritten as the trace $F(H) = \text{Tr}\left[ H \frac{a_0 a_0^T}{N} \right]$ so that we can effectively use the independence of $H$ with $a_0$ and compute the expectation $\mathbb{E}_{a_0}[F(H)] = \frac{r^2}{N}\text{Tr}[H]$. Hence if $F(H)$ concentrates as $N \to \infty$, we can replace it by $\lim_N \frac{r^2}{N}\text{Tr}[H]$.

In the sequel we will adopt the following notation. For any sequence of matrices $(M_k) \in \mathbb{R}^{k \times k}$ we set $\text{Tr}_k[M_k] = \lim_{k \to \infty} \frac{1}{k}\text{Tr}[M_k]$.

Therefore, in general, applying the concentration arguments above, we can substitute the limiting expressions with the following terms

$$L_0(z) = \frac{a_0^T}{\sqrt{N}} R(z) \frac{a_0}{\sqrt{N}} \xrightarrow[d\to\infty]{} r^2 \, \text{Tr}_N[R(z)] \tag{51}$$

$$K(z) = \frac{\beta^T}{\sqrt{d}} \frac{\Theta^T}{\sqrt{d}} R(z) \frac{Z^T Y}{N} \xrightarrow[d\to\infty]{} \text{Tr}_d\left[ \frac{\Theta^T}{\sqrt{d}} R(z) \frac{Z^T}{\sqrt{N}} \frac{X}{\sqrt{N}} \right] \tag{52}$$

$$H_0(x,y) = \frac{a_0^T}{\sqrt{N}} R(x) \frac{\Theta \Theta^T}{d} R(y) \frac{a_0}{\sqrt{N}} \xrightarrow[d\to\infty]{} r^2 \, \text{Tr}_N\left[ R(x) \frac{\Theta \Theta^T}{d} R(y) \right] \tag{53}$$

$$V(z) = \frac{Y^T Z}{N} R(z) \frac{Z^T Y}{N} \xrightarrow[d\to\infty]{} \text{Tr}_d\left[ \frac{X^T}{\sqrt{N}} \frac{Z}{\sqrt{N}} R(z) \frac{Z^T}{\sqrt{N}} \frac{X}{\sqrt{N}} \right] + s^2 \text{Tr}_N\left[ \frac{Z}{\sqrt{N}} R(z) \frac{Z^T}{\sqrt{N}} \right] \tag{54}$$

$$W(x,y) \xrightarrow[d\to\infty]{} \text{Tr}_d\left[ \frac{X^T}{\sqrt{N}} \frac{Z}{\sqrt{N}} R(x) \frac{\Theta \Theta^T}{d} R(y) \frac{Z^T}{\sqrt{N}} \frac{X}{\sqrt{N}} \right] + s^2 \text{Tr}_N\left[ \frac{Z}{\sqrt{N}} R(x) \frac{\Theta \Theta^T}{d} R(y) \frac{Z^T}{\sqrt{N}} \right] \tag{55}$$

As for the training error, all the required terms are given by $V(z), L_0(z), U_0(z)$, of which only $V(z), L_0(z)$ contributes to the result as $d \to \infty$

Finally, we apply the gaussian equivalence principle with the substitution described in 4.2 with the linearization $Z \to Z_{\text{lin}}$ with $Z_{\text{lin}} \equiv \frac{\mu}{\sqrt{d}} X \Theta^T + \nu \Omega$. This substitution is applied throughout all the occurrences of $Z$, including in the resolvents $z \to R(z)$.

# D Linear Pencil

## D.1 Main matrix

The main approach of the linear-pencil method is to design a block-matrix $M_{x,y} = \sum_{i,j} E_{i,j} \otimes M_{x,y}^{(i,j)}$ where the blocks $M_{x,y}^{(i,j)}$ are either a gaussian random matrix or a scalar matrix, and $E_{i,j}$ is the matrix with matrix elements $(E_{i,j})_{k,l} = \delta_{ki}\delta_{lj}$. The subscripts indicate explicitly the dependence on two complex variables $(x, y) \in \mathbb{C}^2$. Importantly, this matrix is inverted using block-inversion formula to have an expression of the form $M_{x,y}^{-1} = \sum_{i,j} E_{i,j} \otimes (M_{x,y}^{-1})^{(i,j)}$ such that some blocks $(M_{x,y}^{-1})^{(i,j)}$ match the different matrix terms in equations (51).

In order to define our main linear pencil matrix, we first need to introduce some additional upper-level blocks: $U^T = [\frac{X}{\sqrt{N}}, \nu \frac{\Omega}{\sqrt{N}}]$ and $V^T = [\mu \frac{\Theta}{\sqrt{d}}, I]$. In addition, in order to keep a consistent symmetry and structure to our block-matrix, we will use the following blocks in reverse order: $\bar{U}^T = [\nu \frac{\Omega}{\sqrt{N}}, \frac{X}{\sqrt{N}}]$ and $\bar{V}^T = [I, \mu \frac{\Theta}{\sqrt{d}}]$. Furthermore, we let $K_x = (-xI + \frac{Z_{\text{lin}}^T Z_{\text{lin}}}{\sqrt{N}})^{-1}$ and $L_x = (-xI + UU^T VV^T)^{-1}$ and $R_x = (-xI + VV^T UU^T)^{-1}$ and $\tilde{K}_x = (-xI + \frac{Z_{\text{lin}} Z_{\text{lin}}^T}{\sqrt{N}})^{-1}$. The following identities (which can be obtained with the push-through identity) provide additional relations which can be used later:

$$\frac{Z_{\text{lin}}}{\sqrt{N}} = U^T V \tag{56}$$

$$L_x UU^T = U \tilde{K}_x U^T \tag{57}$$

$$VV^T L_x = V K_x V^T \tag{58}$$

$$-x\tilde{K}_x = I - \left(-xI + \frac{Z_{\text{lin}} Z_{\text{lin}}^T}{N}\right)^{-1} \frac{Z_{\text{lin}} Z_{\text{lin}}^T}{N} = I - \frac{Z_{\text{lin}}}{\sqrt{N}} K_x \frac{Z_{\text{lin}}^T}{\sqrt{N}} \tag{59}$$

We define our main block-matrix consisting in $13 \times 13$ blocks where the upper-level blocks $U, V, \bar{U}, \bar{V}$ are to be considered as "flattened":

$$M_{x,y} = \left[\begin{array}{cccc||c|cccc}
-xI & -V^T & 0 & 0 & \frac{\Theta}{\sqrt{d}} & 0 & 0 & 0 & 0 \\
\hline
0 & I & U & 0 & 0 & 0 & 0 & 0 & 0 \\
0 & 0 & I & U^T & 0 & 0 & 0 & 0 & 0 \\
V & 0 & 0 & I & 0 & 0 & 0 & 0 & 0 \\
\hline\hline
0 & 0 & 0 & 0 & I & 0 & 0 & 0 & \frac{\Theta^T}{\sqrt{d}} \\
\hline
0 & 0 & 0 & 0 & 0 & I & \bar{U} & 0 & 0 \\
0 & 0 & 0 & 0 & 0 & 0 & I & \bar{U}^T & 0 \\
0 & 0 & 0 & 0 & 0 & 0 & 0 & I & -\bar{V} \\
\hline
0 & 0 & 0 & 0 & 0 & V^T & 0 & 0 & -yI
\end{array}\right] \tag{60}$$

This is precisely the block-matrix $M$ given at the end of Sect. 4.

## D.2 Linear-pencil inversion and relation to the matrix terms

The inverse of $M_{x,y}$ can be computed by splitting it into higher-level blocks. These blocks are highlighted with the lines and double-lines depicted in equation (60): the block-matrix is split into a $2 \times 2$ block-matrix recursively in order to apply the block-matrix inversion formula recursively. Starting with the higher level split:

$$M_{x,y} = \left[\begin{array}{c||c} M_{1,1} & M_{1,2} \\ \hline 0 & M_{2,2} \end{array}\right] \implies M_{x,y}^{-1} = \left[\begin{array}{c||c} M_{1,1}^{-1} & -M_{1,1}^{-1} M_{1,2} M_{2,2}^{-1} \\ \hline 0 & M_{2,2}^{-1} \end{array}\right] \tag{61}$$

It is now quite straightforward algebra to proceed with the remaining blocks. Starting with $M_{1,1}$:

$$M_{1,1}^{-1} = \begin{bmatrix} K_x & K_x V^T & -K_x \frac{Z_{\text{lin}}^T}{\sqrt{N}} & K_x \frac{Z_{\text{lin}}^T}{\sqrt{N}} U^T \\ -U \frac{Z_{\text{lin}}}{\sqrt{N}} K_x & -x L_x & x L_x U & -x L_x U U^T \\ \frac{Z_{\text{lin}}}{\sqrt{N}} K_x & \frac{Z_{\text{lin}}}{\sqrt{N}} V^T L_x & -x \tilde{K}_x & x \tilde{K}_x U^T \\ -V K_x & -V V^T L_x & V \frac{Z_{\text{lin}}^T}{\sqrt{N}} \tilde{K}_x & -x R_x \end{bmatrix} \tag{62}$$

For $M_{2,2}$, with an additional split:

$$M_{2,2} = \left[ \begin{array}{c|c} I & N_{1,2} \\ 0 & N_{2,2} \end{array} \right] \implies M_{2,2}^{-1} = \left[ \begin{array}{c|c} I & -N_{1,2} N_{2,2}^{-1} \\ 0 & N_{2,2}^{-1} \end{array} \right] \tag{63}$$

A straightforward algebra calculation provides the result of $M_{2,2}^{-1}$:

$$M_{2,2}^{-1} = \begin{bmatrix} I & \frac{\Theta^T}{\sqrt{d}} K_y \bar{V}^T & -\frac{\Theta^T}{\sqrt{d}} K_y \frac{Z_{\text{lin}}^T}{\sqrt{N}} & \frac{\Theta^T}{\sqrt{d}} K_y \frac{Z_{\text{lin}}^T}{\sqrt{N}} \bar{U}^T & -\frac{\Theta^T}{\sqrt{d}} K_y \\ 0 & -y \bar{R}_y & y \bar{U} \tilde{K}_y & -y \bar{U} \bar{U}^T \bar{L}_y & \bar{U} \frac{Z_{\text{lin}}}{\sqrt{N}} K_y \\ 0 & \tilde{K}_y \frac{Z_{\text{lin}}}{\sqrt{N}} \bar{V}^T & -y \bar{K}_y & y \bar{U}^T \bar{L}_y & -\frac{Z_{\text{lin}}}{\sqrt{N}} K_y \\ 0 & -\bar{L}_y \bar{V} \bar{V}^T & \bar{L}_y \bar{V} \frac{Z_{\text{lin}}^T}{\sqrt{N}} & -y \bar{L}_y & \bar{V} K_y \\ 0 & -K_y \bar{V}^T & K_y \frac{Z_{\text{lin}}^T}{\sqrt{N}} & -K_y \frac{Z_{\text{lin}}^T}{\sqrt{N}} \bar{U}^T & K_y \end{bmatrix} \tag{64}$$

Finally, using $Q = K_x \frac{\Theta \Theta^T}{d} K_y$ we obtain the third block of $M_{x,y}$:

$$-M_{1,1}^{-1} M_{1,2} M_{2,2}^{-1} = \begin{bmatrix} -K_x \frac{\Theta}{\sqrt{d}} & -Q \bar{V}^T & Q \frac{Z_{\text{lin}}^T}{\sqrt{N}} & -Q \frac{Z_{\text{lin}}^T \bar{U}^T}{\sqrt{N}} & Q \\ \frac{U Z_{\text{lin}}}{\sqrt{N}} K_x \frac{\Theta}{\sqrt{d}} & \frac{U Z_{\text{lin}}}{\sqrt{N}} Q \bar{V}^T & -\frac{U Z_{\text{lin}}}{\sqrt{N}} Q \frac{Z_{\text{lin}}^T}{\sqrt{N}} & \frac{U Z_{\text{lin}}}{\sqrt{N}} Q \frac{Z_{\text{lin}}^T \bar{U}^T}{\sqrt{N}} & -\frac{U Z_{\text{lin}}}{\sqrt{N}} Q \\ -\frac{Z_{\text{lin}}}{\sqrt{N}} K_x \frac{\Theta}{\sqrt{d}} & -\frac{Z_{\text{lin}}}{\sqrt{N}} Q \bar{V}^T & \frac{Z_{\text{lin}}}{\sqrt{N}} Q \frac{Z_{\text{lin}}^T}{\sqrt{N}} & -\frac{Z_{\text{lin}}}{\sqrt{N}} Q \frac{Z_{\text{lin}}^T \bar{U}^T}{\sqrt{N}} & \frac{Z_{\text{lin}}}{\sqrt{N}} Q \\ V K_x \frac{\Theta}{\sqrt{d}} & V Q \bar{V}^T & -V Q \frac{Z_{\text{lin}}^T}{\sqrt{N}} & V Q \frac{Z_{\text{lin}}^T \bar{U}^T}{\sqrt{N}} & -V Q \end{bmatrix} \tag{65}$$

Notice now that all the matrix terms in equations (51) are actually contained in some of the blocks of our matrix (note that $\text{Tr}_d \left[ \frac{X^T X}{n} \right] = 1$):

$$\bar{L}_0(y) = r^2 \text{Tr}_N \left[ K_y \right] \tag{66}$$

$$\bar{K}(y) = \text{Tr}_d \left[ \frac{\Theta^T}{\sqrt{d}} K_y \frac{Z_{\text{lin}}^T}{\sqrt{N}} \bar{U}^T \right]_{1,2} \tag{67}$$

$$\bar{H}_0(x,y) = r^2 \text{Tr}_N \left[ Q \right] \tag{68}$$

$$\bar{W}(x,y) = s^2 \frac{\phi}{\psi} \text{Tr}_n \left[ \frac{Z_{\text{lin}}}{\sqrt{N}} Q \frac{Z_{\text{lin}}^T}{\sqrt{N}} \right] + \text{Tr}_d \left[ \frac{U Z_{\text{lin}}}{\sqrt{N}} Q \frac{Z_{\text{lin}}^T \bar{U}^T}{\sqrt{N}} \right]_{1,2} \tag{69}$$

$$\bar{V}(x) = s^2 \frac{\phi}{\psi} \text{Tr}_n \left[ I_n + x \tilde{K}_x \right] + \left( \text{Tr}_d \left[ x L_x U U^T \right]_{1,1} + \text{Tr}_d \left[ \frac{X^T X}{N} \right] \right) \tag{70}$$

Or equivalently, with the block coordinates of the inverse matrix $M_{x,y}^{-1}$:

$$\bar{L}_0(y) = r^2 \text{Tr}_N \left[ (M_{x,y}^{-1})^{(13,13)} \right] \tag{71}$$

$$\bar{K}(y) = \text{Tr}_d \left[ (M_{x,y}^{-1})^{(7,12)} \right] \tag{72}$$

$$\bar{H}_0(x,y) = r^2 \text{Tr}_N \left[ (M_{x,y}^{-1})^{(1,13)} \right] \tag{73}$$

$$\bar{W}(x,y) = s^2 \frac{\phi}{\psi} \text{Tr}_n \left[ (M_{x,y}^{-1})^{(4,10)} \right] + \text{Tr}_d \left[ (M_{x,y}^{-1})^{(2,12)} \right] \tag{74}$$

$$\bar{V}(x) = s^2 \frac{\phi}{\psi} \left( 1 - \text{Tr}_n \left[ (M_{x,y}^{-1})^{(4,4)} \right] \right) + \left( -\text{Tr}_d \left[ (M_{x,y}^{-1})^{(2,5)} \right] + \frac{\phi}{\psi} \right) \tag{75}$$

In the next section we show how to derive further each trace of the squared matrices from the block matrix $M_{x,y}$. In order to deal with self-adjoint matrices, we double the dimensions with $\tilde{M}_{x,y}$:

$$\tilde{M}_{x,y} = \begin{bmatrix} 0 & M_{x,y} \\ M_{x,y}^{\dagger} & 0 \end{bmatrix} \tag{76}$$

and find the inverse:

$$\tilde{M}_{x,y}^{-1} = \begin{bmatrix} 0 & (M_{x,y}^{\dagger})^{-1} \\ M_{x,y}^{-1} & 0 \end{bmatrix} \tag{77}$$

### D.3  Structural terms of the limiting traces

The matrix $M_{x,y}$ is a block-matrix constituted with either gaussian random matrices, or constant matrices (proportional to $I$). More precisely, letting $S$ be the matrix of the coefficients of the constant blocks of $M_{x,y}$ (and $\tilde{S}$ for $\tilde{M}_{x,y}$), and $A$ the random blocks part ($\tilde{A}$ respectively) we write : $\tilde{M}_{x,y} = \sum_{i,j} E_{i,j} \otimes \tilde{M}_{x,y}^{(i,j)}$ where $\tilde{M}_{x,y}^{(i,j)} = \tilde{S}^{(i,j)} + \tilde{A}^{(i,j)}$ is the block of size $(N_i, N_j)$. Also notice that letting $\mathbb{L} = \{(i,j)|\ N_i = N_j\}$, the fact that the constant blocks are supposed to be proportional to an identity matrix implies that: $\forall (i,j) \notin \mathbb{L} \implies \tilde{S}^{(i,j)} = 0 = z_{i,j} 0_{N_i, N_j}$ with $0_{N_i, N_j}$ the zero-matrix of size $N_i \times N_j$ and otherwise $\forall (i,j) \in \mathbb{L} \implies \tilde{S}^{(i,j)} = z_{i,j} I_{N_i}$ with $\tilde{B} = (z_{i,j})$ the matrix of size $26 \times 26$.

Now we want to find a matrix $\tilde{G} \in \mathbb{R}^{26 \times 26}$ such that

$$[\tilde{G}]_{i,j} = \text{Tr}_{N_i}\left[(\tilde{M}_{x,y}^{-1})^{(i,j)}\right], \quad \forall (i,j) \in \mathbb{L}, \tag{78}$$

An important theorem in [38] (chapter 9, equ. (9.5) and theorem 2), which we show again in the next section, states that there is a solution $\tilde{G}$ of the equation

$$\tilde{B}\tilde{G} = I + \eta(\tilde{G})\tilde{G} \tag{79}$$

which satisfies (78). In this equation $\eta(\tilde{G})$ is the matrix mapping defined element-wise as:

$$[\eta(\tilde{G})]_{i,j} = \delta_{\mathbb{L}}(i,j) \cdot \sum_{k,l \in \mathbb{L}} \sigma(i,k;l,j) \cdot [\tilde{G}]_{k,l} \tag{80}$$

and where $\sigma$ satisfies the relation for all $(i,k,l,j)$ such that $N_i = N_j$ and $N_k = N_l$ (and keeping in mind that the $N_k$ are growing with the dimension $d$):

$$\forall (r,s) \in \{1,\ldots,N_i\} \times \{1,\ldots,N_j\}, r \neq s \implies \sigma(i,k;l,j) = \lim_{d \to \infty} N_k \cdot \mathbb{E}\left[[\tilde{A}^{(i,k)}]_{r,s}[\tilde{A}^{(l,j)}]_{s,r}\right] \tag{81}$$

We remark that the setting here, and in particular equation (79), is in fact more general than in [38] (chapter 9, equ. (9.5)) and we provide an independent and self-contained (formal) derivation of (79) in Appendix E using the replica method.

For example, we have $M_{x,y}^{(5,1)} = \mu \frac{\Theta^T}{\sqrt{d}}$ of size $d \times N$ and $M_{x,y}^{(1,7)} = \frac{\Theta}{\sqrt{d}}$ of size $N \times d$. So this is $\tilde{M}_{x,y}^{(5,14)} = \mu \frac{\Theta^T}{\sqrt{d}}$ and $\tilde{M}_{x,y}^{(1,20)} = \frac{\Theta}{\sqrt{d}}$, with $N_5 = N_{20} = d$ and $N_{14} = N_1 = N$. For $r = 1, s = 2$ (or any other suitable indices) we find:

$$\sigma(5,14;1,20) = \lim_{d \to \infty} \mu \frac{N}{d} \mathbb{E}\left[[\Theta]_{1,2}^2\right] = \mu \psi$$

In fact, a careful inspection of all the blocks in row 5 and all the blocks in column 20 shows that we have $[\eta(\tilde{G})]_{5,20} = \mu \psi [\tilde{G}]_{14,1}$.

Calculating all the terms of $\eta(\tilde{G})$ is quite cumbersome, but it can be done automatically with the help of a computer algebra system. Still, this approach yields many equations for each $26 \times 26$ terms of $\tilde{G}$. However, some initial structure can also be provided for this matrix. Looking back at $\tilde{M}_{x,y}^{-1}$, it is clear that some blocks will have the same limiting traces (potentially seen using the aforementioned push-through identities). For instance, $(M_{1,1}^{-1})^{(1,1)} = K_x = -(M_{1,1}^{-1})^{(6,1)}$ (expanding the $U, V$ blocks), so $(M_{x,y}^{-1})^{(1,1)} = -(M_{x,y}^{-1})^{(6,1)}$, in other words $(\tilde{M}_{x,y}^{-1})^{(14,1)} = -(\tilde{M}_{x,y}^{-1})^{(19,1)}$, and thus we

expect $[\tilde{G}]_{14,1} = -[\tilde{G}]_{19,1}$. Non-squared blocks can also be mapped to $0$ in $\tilde{G}$. In the end, taking every block into account, $\tilde{G}$ is expected to be of the form:

$$\tilde{G} = \left[\begin{array}{c|c} 0 & G^\dagger \\ \hline G & 0 \end{array}\right] \tag{82}$$

with

$$G = \left[\begin{array}{c|c|c} G_{1,1} & G_{1,2} & G_{1,3} \\ \hline 0 & 1 & G_{2,3} \\ \hline 0 & 0 & G_{3,3} \end{array}\right] \tag{83}$$

(which has $13 \times 13$ scalar matrix elements) where:

$$G_{1,3} = \left[\begin{array}{cc|cc|cc} -q_1 & 0 & 0 & -\nu q_6^{yx} & 0 & q_1 \\ 0 & \mu q_7^{yx} & 0 & 0 & q_2 & 0 \\ \nu q_6^{xy} & 0 & 0 & \nu^2 q_3 & 0 & -\nu q_6^{xy} \\ 0 & 0 & q_4 & 0 & 0 & 0 \\ 0 & \mu^2 q_5 & 0 & 0 & \mu q_7^{xy} & 0 \\ q_1 & 0 & 0 & \nu q_6^{yx} & 0 & -q_1 \end{array}\right] \tag{84}$$

$$G_{1,1} = \left[\begin{array}{c|cc|cc|c} g_1^x & 0 & g_1^x & 0 & 0 & \nu g_2^x \\ 0 & h_1^x & 0 & 0 & h_4^x & 0 \\ -\nu g_2^x & 0 & h_2^x & 0 & 0 & \nu^2 h_5^x \\ 0 & 0 & 0 & g_3^x & 0 & 0 \\ 0 & -\mu^2 h_3^x & 0 & 0 & h_1^x & 0 \\ -g_1^x & 0 & -g_1^x & 0 & 0 & h_2^x \end{array}\right] \tag{85}$$

$$G_{3,3} = \left[\begin{array}{cc|c|cc|c} h_2^y & 0 & 0 & \nu^2 h_5^y & 0 & \nu g_2^y \\ 0 & h_1^y & 0 & 0 & h_4^y & 0 \\ 0 & 0 & g_3^y & 0 & 0 & 0 \\ -g_1^y & 0 & 0 & h_2^y & 0 & g_1^y \\ 0 & -\mu^2 h_3^y & 0 & 0 & h_1^y & 0 \\ -g_1^y & 0 & 0 & -\nu g_2^y & 0 & g_1^y \end{array}\right] \tag{86}$$

$$G_{1,2} = \left[\begin{array}{c} 0 \\ \hline t_1^x \\ 0 \\ \hline 0 \\ \hline \mu h_3^x \\ 0 \end{array}\right] \qquad G_{2,3} = \left[\begin{array}{c|c|c|c|c|c} 0 & \mu h_3^y & 0 & 0 & t_1^y & 0 \end{array}\right] \tag{87}$$

All (non-vanishing) matrix elements depend on the complex variables $x$ and $y$. This is indicated by the upper-script notation with $x, y, xy, yx$. Some quantities depend only on $x$, some only on $y$, and some on both $x$ and $y$. Among the ones that depend on both variables the quantities $q_6^{xy}, q_6^{yx}, q_7^{xy}, q_7^{yx}$ are *non-symmetric*, while $q_1, q_2, q_3, q_4, q_5$ are *symmetric* (e.g., $q_1^{x,y} = q_1^{y,x}$). We choose not to use the upper-script notation for the symmetric quantities in order to distinguish them from the *non-symmetric* ones.

Eventually, with a careful mapping between $\tilde{M}_{x,y}^{-1}$ and $\tilde{G}$ in equations (66), only $g_1^x, t_1^x, h_4^x, g_3^x$ and the symmetric terms $q_1, q_2, q_4$ are needed and equations (66) take the form:

$$\bar{L}_0(x) = r^2 g_1^x \tag{88}$$
$$\bar{K}(x) = t_1^x \tag{89}$$
$$\bar{H}_0(x,y) = r^2 q_1 \tag{90}$$
$$\bar{W}(x,y) = s^2 \frac{\phi}{\psi} q_4 + q_2 \tag{91}$$
$$\bar{V}(x) = s^2 \frac{\phi}{\psi}(1 - g_3^x) + \left(\frac{\phi}{\psi} - h_4^x\right) \tag{92}$$

## D.4 Solution of the fixed point equation

The fixed-point equations as described in (79) for the given matrices $\tilde{S}, \eta(\tilde{G}), \tilde{G}$ is a priori a system of $26 \times 26$ algebraic equations. are computed using Sympy in python, a symbolic calculation tool. In effect this is really a fixed point equation for $G$ a priori involving $13 \times 13$ algebraic equations. It turns out that many matrix elements vanish and (using the symbolic calculation tool Sympy in python) we can extract a system of 39 algebraic equations which are given in the following:

$$0 = g_1^x \left( -\mu^2 h_4^x + x \right) - g_2^x \nu + 1 \tag{93}$$

$$0 = g_1^x \left( -\mu^2 h_4^x + x \right) + h_2^x \tag{94}$$

$$0 = g_2^x \nu \left( -\mu^2 h_4^x + x \right) + h_5^x \nu^2 \tag{95}$$

$$0 = -g_1^y \left( \mu^2 q_2 - \mu t_1^x - \mu t_1^y + 1 \right) + \nu q_6^{xy} - q_1 \left( -\mu^2 h_4^x + x \right) \tag{96}$$

$$0 = -g_2^y \nu \left( \mu^2 q_2 - \mu t_1^x - \mu t_1^y + 1 \right) + \nu^2 q_3 - \nu q_6^{yx} \left( -\mu^2 h_4^x + x \right) \tag{97}$$

$$0 = g_1^y \left( \mu^2 q_2 - \mu t_1^x - \mu t_1^y + 1 \right) - \nu q_6^{xy} + q_1 \left( -\mu^2 h_4^x + x \right) \tag{98}$$

$$0 = \frac{\mu^2 \phi g_3^x h_3^x}{\psi} - h_1^x + 1 \tag{99}$$

$$0 = \frac{\phi g_3^x h_1^x}{\psi} - h_4^x \tag{100}$$

$$0 = -\frac{\mu \phi g_3^x h_3^x}{\psi} - t_1^x \tag{101}$$

$$0 = \frac{\mu^2 \phi g_3^x q_5}{\psi} + \frac{\mu^2 \phi h_3^y q_4}{\psi} - \mu q_7^{yx} \tag{102}$$

$$0 = \frac{\mu \phi g_3^x q_7^{xy}}{\psi} + \frac{\phi h_1^y q_4}{\psi} - q_2 \tag{103}$$

$$0 = -\frac{\phi g_1^x g_3^x \nu^2}{\psi} + g_2^x \nu \tag{104}$$

$$0 = -\frac{\phi g_1^x g_3^x \nu^2}{\psi} - h_2^x + 1 \tag{105}$$

$$0 = \frac{\phi g_3^x h_2^x \nu^2}{\psi} - h_5^x \nu^2 \tag{106}$$

$$0 = -\frac{\phi g_1^y \nu^2 q_4}{\psi} + \frac{\phi g_3^x \nu^2 q_1}{\psi} - \nu q_6^{xy} \tag{107}$$

$$0 = \frac{\phi g_3^x \nu^3 q_6^{yx}}{\psi} + \frac{\phi h_2^y \nu^2 q_4}{\psi} - \nu^2 q_3 \tag{108}$$

$$0 = \frac{\phi g_1^y \nu^2 q_4}{\psi} - \frac{\phi g_3^x \nu^2 q_1}{\psi} + \nu q_6^{xy} \tag{109}$$

$$0 = g_3^x \left( \frac{\mu^2 h_3^x}{\psi} - g_1^x \nu^2 - 1 \right) + 1 \tag{110}$$

$$0 = g_3^y \left( \frac{\mu^2 q_5}{\psi} + \nu^2 q_1 \right) + q_4 \left( \frac{\mu^2 h_3^x}{\psi} - g_1^x \nu^2 - 1 \right) \tag{111}$$

$$0 = -\mu^2 \psi g_1^x h_1^x - \mu^2 h_3^x \tag{112}$$

$$0 = -\mu^2 \psi g_1^x h_4^x - h_1^x + 1 \tag{113}$$

$$0 = -\mu^2 \psi g_1^x t_1^x + \mu \psi g_1^x + \mu h_3^x \tag{114}$$

$$0 = -\mu^3 \psi g_1^x q_7^{yx} - \mu^2 \psi g_1^x h_3^y + \mu^2 \psi h_1^y q_1 - \mu^2 q_5 \tag{115}$$

$$0 = -\mu^2 \psi g_1^x q_2 + \mu^2 \psi h_4^y q_1 + \mu \psi g_1^x t_1^y - \mu q_7^{xy} \tag{116}$$

$$0 = -g_2^x \nu - h_2^x + 1 \tag{117}$$

$$0 = \mu\psi g_1^y h_1^y + \mu h_3^y \tag{118}$$

$$0 = \mu\psi g_1^y h_4^y - t_1^y \tag{119}$$

$$0 = -\frac{\phi g_1^y g_3^y \nu^2}{\psi} - h_2^y + 1 \tag{120}$$

$$0 = \frac{\phi g_3^y h_2^y \nu^2}{\psi} - h_5^y \nu^2 \tag{121}$$

$$0 = \frac{\phi g_1^y g_3^y \nu^2}{\psi} - g_2^y \nu \tag{122}$$

$$0 = \frac{\mu^2 \phi g_3^y h_3^y}{\psi} - h_1^y + 1 \tag{123}$$

$$0 = \frac{\phi g_3^y h_1^y}{\psi} - h_4^y \tag{124}$$

$$0 = g_3^y \left( \frac{\mu^2 h_3^y}{\psi} - g_1^y \nu^2 - 1 \right) + 1 \tag{125}$$

$$0 = -g_2^y \nu - h_2^y + 1 \tag{126}$$

$$0 = -\mu^2 \psi g_1^y h_1^y - \mu^2 h_3^y \tag{127}$$

$$0 = -\mu^2 \psi g_1^y h_4^y - h_1^y + 1 \tag{128}$$

$$0 = -g_1^y \left( -\mu^2 h_4^y + y \right) - h_2^y \tag{129}$$

$$0 = -g_2^y \nu \left( -\mu^2 h_4^y + y \right) - h_5^y \nu^2 \tag{130}$$

$$0 = g_1^y \left( -\mu^2 h_4^y + y \right) - g_2^y \nu + 1 \tag{131}$$

### D.5 Reduction of the solutions

The previous system of equations can be reduced further by substitutions with a computer algebra system. We find the variables $g_3^x, t_1^x, h_4^x, g_1^x, h_1^x$ are linked through the algebraic system:

$$\begin{cases} 0 = 1 + g_1^x \left( -\mu^2 h_4^x - \frac{\phi}{\psi} g_3^x u^2 + x \right) \\ 0 = -h_4^x + g_3^x \left( -\mu^2 \phi g_1^x h_4^x + \frac{\phi}{\psi} \right) \\ 0 = \frac{\phi}{\psi}(1 - g_3^x) - g_1^x x - 1 \\ 0 = \mu\psi g_1^x h_4^x - t_1^x \\ 0 = 1 - h_1^x - \mu t_1^x \end{cases} \tag{132}$$

Notice this system can be shrinked further down to 3 equations to get to the main result in 3.1 using the substitution $h_1^x$ with the $5^{\text{th}}$ equation and $g_3^x$ with the $3^{\text{rd}}$ equation. Also, by symmetry we find the same equations for $g_3^y, t_1^y, h_4^y, g_1^y, h_1^y$.

For the other variables, a set of equations link $q_1, q_2, q_4, q_5$. Notice there can many different representations depending on the reductions that are applied. Here we only show the example which has been used throughout the computations:

$$\begin{cases} 0 = -\mu^2 g_1^y q_2 + \mu^2 h_4^x q_1 + \mu g_1^y t_1^x + \mu g_1^y t_1^y - \frac{\phi g_1^y q_4 \nu^2}{\psi} - g_1^y - q_1 x + \frac{q_1 \nu^2 (\phi - \psi g_1^x x - \psi)}{\psi} \\ 0 = \mu \left( \phi - \psi g_1^x x - \psi \right) \left( -\mu g_1^x q_2 + \mu h_4^y q_1 + g_1^x t_1^y \right) + \frac{\phi h_1^y q_4}{\psi} - q_2 \\ 0 = -\mu^2 g_1^x h_1^x q_4 + \frac{\mu^2 q_5 \left( \phi - \psi g_1^y y - \psi \right)}{\phi\psi} - g_1^x q_4 \nu^2 - q_4 + \frac{q_1 \nu^2 \left( \phi - \psi g_1^y y - \psi \right)}{\phi} \\ 0 = \mu^2 \phi g_1^x g_1^y h_1^y q_4 - \frac{\mu^2 g_1^x q_5 (\phi - \psi g_1^x x - \psi)}{\psi} + \psi g_1^x g_1^y h_1^y + h_1^y q_1 - \frac{q_5}{\psi} \end{cases} \tag{133}$$

In conclusion, we can obtain 3 systems with $(4, 5, 5)$-equations or 3 systems with $(4, 3, 3)$-equations (so a total of 10), as in the main result 3.1 (as discussed above these various systems are all equivalent and depend on the applied reductions).

The solutions are not necessarily unique and one has to choose the appropriate ones with care. In our experimental results using Matlab with the "vpasolve" function, conditioning on $\operatorname{Im} g_1^x > 0$ and $\operatorname{Im} g_3^x > 0$ provided a unique solution to (132) for $x \in \mathbb{R}_+$ (or $x \in \mathbb{R} \times i[0, \epsilon]$ for $\epsilon$ close to 0 ); while

conditioning on $g_1^x, g_3^x \in \mathbb{R}_+$ provided a unique solution to (132) for $x \in \mathbb{R}_-$. We remind that we use $x \in \mathbb{R}_-$ exclusively in the time limit $t \to \infty$ in result 3.2 while we use $x \in \mathbb{R}_+$ in the situation of result 3.1. In addition, we found that selecting the appropriate solutions for $x$ and $y$ as just described for (132) also led to a unique solution for 133 in our experiments.

# E  Linear pencil method from the replica trick argument

A general approach to solve random matrix problems is to use the replica method, and historically this goes back to [39]. In this appendix we show how to derive the fixed point equation (79) and (81) as well as (78) in appendix D. Such equations have been rigorously proved thanks to combinatorial methods in the recent literature on random matrix theory (see [38], chapter 9, equ. (9.5)), but here we give a self-contained derivation using the replica trick, similar in spirit to [39]. Although our derivation is far from rigorous it does covers linear pencils with a more general structure than in [38], chapter 9, which are needed for our purposes.

**Setting the replica calculation.**  Let $(N_1, \ldots, N_d) \in \mathbb{N}^d$ for some $d \in \mathbb{N}$, and $N = \sum_{i=1}^d N_i$. and let's consider a symmetric[2] block matrix, called the "linear pencil", with $M = \sum_{i,j} E_{i,j} \otimes M^{(i,j)}$ such that $M^{(i,j)}$ is a matrix of size $N_i \times N_j$ and $E_{i,j}$ the matrix with elements $(E_{i,j})_{kl} = \delta_{ki}\delta_{lj}$. We assume that we can decompose $M = R + S$ with two block-matrices $R$ and $S$ such that the blocks $R^{(i,j)}$ are sum of independent real gaussian random matrices (with possibly their transpose) and $S^{(i,j)} = 0$ if $N_i \neq N_j$ or a scalar matrix $S^{(i,j)} = z_{i,j} \cdot I_{N_i}$ if $N_i = N_j$ (with $I_{N_i}$ the $N_i \times N_i$ identity). Given the list of squares-blocks $\mathbb{L} = \{i, j | N_i = N_j\}$, we let $B = (z_{i,j})$ be the matrix[3] of the scalar coefficients in $\mathbb{C}$ where it is assumed $z_{i,j} = 0$ when $(i, j) \notin \mathbb{L}$.

Now we have defined a standard linear pencil. As a side remark, note also that we can accomodate random blocks $R^{(i,j)}$ which are *symmetric* gaussian random matrices (i.e., the lower and upper triangular parts are not independent) because we can always decompose them into the sum of two random matrices, $R^{(i,j)} = Y + Y^\mathsf{T}$.

In general, let $\{Y_k\}_{1 \le k \le K}$ be a list of $K$ independent gaussian random matrices with i.i.d elements of variance $N^{-1}$, and various heights and widths among $\{N_i \times N_j, i, j \in \{1, \cdots, d\}\}$. These will constitute the random blocks of $R^{(i,j)}$ as follows. Given $i, j$ we define the set $\mathbb{S}_{i,j} = \{k | \text{width}(Y_k) = N_i, \text{height}(Y_k) = N_j\}$. We have for some coefficients $\alpha_{i,j}^k$,

$$R = \sum_{i,j=1}^d \sum_{k \in \mathbb{S}_{i,j}} \alpha_{i,j}^k (E_{i,j} \otimes Y_k + E_{j,i} \otimes Y_k^\mathsf{T}). \tag{134}$$

Notice $\alpha_{i,j}^k$ is not necessarily symmetric under exchange of $i$ and $j$, but $R = R^\mathsf{T}$ is still guaranteed to be symmetric (however, if $\alpha_{i,j}^k$ is symmetric for all $k$, then it implies all the random blocks are themselves symmetric). Similarly, we have the scalar block,

$$S = \sum_{(i,j) \in \mathbb{L}}^d z_{i,j} (E_{i,j} \otimes I_{N_i}). \tag{135}$$

Now, let's define $I = \mathbb{E}_{\{Y_k\}} \log \det(M)$. Then using the replica trick,

$$I = -2\mathbb{E} \log \left[\det(M)^{-\frac{1}{2}}\right] \simeq \lim_{n \to 0} 2\mathbb{E} \left[\frac{1 - \det(M)^{-\frac{n}{2}}}{n}\right]. \tag{136}$$

We will first compute the term $J = \mathbb{E} \det(M)^{-\frac{n}{2}}$ for integers $n \ge 1$.

---

[2]Here $M$ plays the role of the symmetric matrix $\tilde{M}$ in appendix D. We remove the tilde to alleviate the notation as this will not create any confusion here

[3]This plays the role of $\tilde{B}$ in appendix D.

**Calculation of $J$ for integer $n \geq 1$.** We start with the Gaussian representation of the determinant

$$J = \mathbb{E}\left[ \prod_{a=1}^{n} \int_{\mathbb{R}^N} \frac{\mathrm{d}x}{(2\pi)^{\frac{N}{2}}} \exp\left( -\frac{1}{2} x^\intercal M x \right) \right]. \tag{137}$$

We have

$$J = \int_{\mathbb{R}^{n \times N}} \exp\left( -\frac{1}{2} \sum_{a=1}^{n} x^{a\intercal} S x^a \right) \mathbb{E}\left[ \exp\left( -\frac{1}{2} \sum_{a=1}^{n} x^{a\intercal} R x^a \right) \right] \prod_{a=1}^{n} \frac{\mathrm{d}x^a}{(2\pi)^{\frac{N}{2}}} \tag{138}$$

where $a = 1, \ldots, n$ is called the "replica index". Setting $x^\intercal = [x_1 | \ldots | x_d]$ where each $x_i$ is of size $N_i$,

$$x^\intercal R x = \sum_{i,j=1}^{d} \left[ x_i^\intercal \left( \sum_{k \in \mathbb{S}_{i,j}} \alpha_{i,j}^k Y_k \right) x_j + x_j^\intercal \left( \sum_{k \in \mathbb{S}_{i,j}} \alpha_{i,j}^k Y_k^\intercal \right) x_i \right]. \tag{139}$$

Then defining the set $\mathbb{S}_k^{-1} = \{(i,j) | N_i = \text{width}(Y_k), N_j = \text{height}(Y_k)\}$ (for a given $k$)

$$x^\intercal R x = \sum_{k=1}^{K} \sum_{(i,j) \in \mathbb{S}_k^{-1}} 2\alpha_{i,j}^k (x_i^\intercal Y_k x_j) \tag{140}$$

an expanding the inner products we can further write down

$$x^\intercal R x = 2 \sum_{k=1}^{K} \sum_{r,s} [Y_k]_{r,s} \sum_{(i,j) \in \mathbb{S}_k^{-1}} \alpha_{i,j}^k [x_i]_r [x_j]_s \tag{141}$$

Thus

$$\mathbb{E}\left[ \exp\left( -\frac{1}{2} \sum_{a=1}^{n} x^{a\intercal} R x^a \right) \right] = \prod_{k,r,s} \mathbb{E}\left[ \exp\left( -[Y_k]_{r,s} \sum_{\substack{1 \leq a \leq n \\ (i,j) \in \mathbb{S}_k^{-1}}} \alpha_{i,j}^k [x_i^a]_r [x_j^a]_s \right) \right] \tag{142}$$

and using the moment generating function of the normal distribution, with $\mathbb{E}[Y_k]_{r,s}^2 = \frac{1}{N}$, we obtain

$$\mathbb{E}\left[ \exp\left( -\frac{1}{2} \sum_{a=1}^{n} x^{a\intercal} R x^a \right) \right] = \exp\left( \frac{1}{2N} \sum_{k,r,s} \left( \sum_{\substack{1 \leq a \leq n \\ (i,j) \in \mathbb{S}_k^{-1}}} \alpha_{i,j}^k [x_i^a]_r [x_j^a]_s \right)^2 \right). \tag{143}$$

But now, we can expand the square using the set $\mathbb{T} = \{(i,j,k,l) \in \{1,\ldots,d\}^4 | (N_i, N_j) = (N_k, N_l)\}$:

$$\sum_{k,r,s} \left( \sum_{\substack{1 \leq a \leq n \\ (i,j) \in \mathbb{S}_k^{-1}}} \alpha_{i,j}^k [x_i^a]_r [x_j^a]_s \right)^2 = \sum_{\substack{1 \leq a \leq n \\ 1 \leq b \leq n}} \sum_{\substack{i_a, j_a, i_b, j_b \\ \in \mathbb{T}}} (x_{i_a}^a \cdot x_{i_b}^b)(x_{j_a}^a \cdot x_{j_b}^b) \sum_{k \in \mathbb{S}_{i_a, j_a}} \alpha_{i_a, j_a}^k \alpha_{i_b, j_b}^k. \tag{144}$$

Notice the symmetry with the indices

$$\sum_{\substack{i_a, j_a, i_b, j_b \\ \in \mathbb{T}}} (x_{i_a}^a \cdot x_{i_b}^b)(x_{j_a}^a \cdot x_{j_b}^b) \sum_{k \in \mathbb{S}_{i_a, j_a}} \alpha_{i_a, j_a}^k \alpha_{i_b, j_b}^k = \sum_{\substack{j_a, i_a, j_b, i_b \\ \in \mathbb{T}}} (x_{i_a}^a \cdot x_{i_b}^b)(x_{j_a}^a \cdot x_{j_b}^b) \sum_{k \in \mathbb{S}_{j_a, i_a}} \alpha_{j_a, i_a}^k \alpha_{j_b, i_b}^k. \tag{145}$$

Therefore, defining $\sigma_{i,j}^{l,k} \equiv \sum_{t \in \mathbb{S}_{i,j}} \alpha_{i,j}^t \alpha_{k,l}^t + \sum_{t \in \mathbb{S}_{j,i}} \alpha_{j,i}^t \alpha_{l,k}^t$

$$\sum_{\substack{i_a, j_a, i_b, j_b \\ \in \mathbb{T}}} (x_{i_a}^a \cdot x_{i_b}^b)(x_{j_a}^a \cdot x_{j_b}^b) \sum_{k \in \mathbb{S}_{i_a, j_a}} \alpha_{i_a, j_a}^k \alpha_{i_b, j_b}^k = \frac{1}{2} \sum_{\substack{i_a, j_a, i_b, j_b \\ \in \mathbb{T}}} (x_{i_a}^a \cdot x_{i_b}^b)(x_{j_a}^a \cdot x_{j_b}^b) \sigma_{i_a, j_a}^{j_b, i_b}. \tag{146}$$

We remark for further use the symmetry property $\sigma_{i,j}^{l,k} = \sigma_{j,i}^{k,l}$.

Now, notice also that we have $x^\intercal B x = \sum_{(i,j)\in\mathbb{L}} z_{i,j}(x_i \cdot x_j)$, so

$$J = \int_{\mathbb{R}^{n\times N}} \exp\left(-\frac{1}{2}\sum_{\substack{1\leq a\leq n\\(i,j)\in\mathbb{L}}} z_{i,j}(x_i^a\cdot x_j^a) + \frac{1}{4N}\sum_{\substack{1\leq a\leq n\\1\leq b\leq n}}\sum_{\substack{i_a,j_a,i_b,j_b\\\in\mathbb{T}}} (x_{i_a}^a\cdot x_{i_b}^b)(x_{j_a}^a\cdot x_{j_b}^b)\sigma_{i_a,j_a}^{j_b,i_b}\right)\prod_{a=1}^n \frac{\mathrm{d}x^a}{(2\pi)^{\frac{N}{2}}} \tag{147}$$

Now, let's define the "overlaps" $q_{i,j}^{a,b} = \frac{1}{N}x_i^a\cdot x_j^b$ for $i,j\in\mathbb{L}$ and $1\leq a,b\leq n$, and 0 otherwise, then

$$J = \int_q\int_{\mathbb{R}^{n\times N}} \left(\prod_{a=1}^n \frac{\mathrm{d}x^a}{(2\pi)^{\frac{N}{2}}}\right)\left(\prod_{\substack{(i,j)\in\mathbb{L}\\1\leq a\leq b\leq n}} \mathrm{d}q_{i,j}^{a,b}\delta\left(q_{i,j}^{a,b} - \frac{x_i^a\cdot x_i^b}{N}\right)\right) e^{N\Xi(q)} \tag{148}$$

with

$$\Xi(q) = -\frac{1}{2}\sum_{a=1}^n\sum_{(i,j)\in\mathbb{L}} z_{i,j}q_{i,j}^{a,a} + \frac{1}{4}\sum_{\substack{1\leq a\leq n\\1\leq b\leq n}}\sum_{\substack{i_a,j_a,i_b,j_b\\\in\mathbb{T}}} q_{i_a,i_b}^{a,b}q_{j_a,j_b}^{a,b}\sigma_{i_a,j_a}^{j_b,i_b}. \tag{149}$$

Eventually, with a Fourier transform representation of the Dirac distribution and a change of variable to have real integrands

$$J = \int_{\hat{q}}\int_q\int_{\mathbb{R}^{n\times N}} \left(\prod_{a=1}^n \frac{\mathrm{d}x^a}{(2\pi)^{\frac{N}{2}}}\right)\left(\prod_{\substack{(i,j)\in\mathbb{L}\\1\leq a\leq b\leq n}} \mathrm{d}q_{i,j}^{a,b}\mathrm{d}\hat{q}_{i,j}^{a,b}\frac{N}{2\pi i}e^{-\hat{q}_{i,j}^{a,b}(Nq_{i,j}^{a,b}-x_i^a\cdot x_j^b)}\right) e^{N\Xi(q)}. \tag{150}$$

So for some constant $C$ we have (the constant turns out to be unimportant in the high-dimensional limit)

$$J = C\int_{\hat{q}}\int_q \left(\prod_{\substack{i,j\\1\leq a\leq b\leq n}} \mathrm{d}q_{i,j}^{a,b}\mathrm{d}\hat{q}_{i,j}^{a,b}\right) e^{N(\Gamma(q,\hat{q})+\Xi(q))+\psi(\hat{q})} \tag{151}$$

where

$$\Gamma(q,\hat{q}) = -\sum_{(i,j)\in\mathbb{L}}\sum_{1\leq a\leq b\leq n} \hat{q}_{i,j}^{a,b}q_{i,j}^{a,b} \tag{152}$$

and

$$\psi(\hat{q}) = \frac{1}{N}\log\int_{\mathbb{R}^{n\times N}} \left(\prod_{\substack{1\leq a\leq n\\1\leq k\leq d}} \frac{\mathrm{d}x_k^a}{(2\pi)^{\frac{N_k}{2}}}\right) e^{\sum_{(i,j)\in\mathbb{L}}\sum_{1\leq a\leq b\leq n}\hat{q}_{i,j}^{a,b}(x_i^a\cdot x_j^b)}. \tag{153}$$

Notice that for $\psi(\hat{q})$ we can further expand the terms over components of $x_k^a = [x_k^a]_{r=1}^{N_k}$

$$\psi(\hat{q}) = \frac{1}{N}\log\int_{\mathbb{R}^{n\times N}} \left(\prod_{\substack{1\leq a\leq n\\1\leq k\leq d}}\prod_{r=1}^{N_k} \frac{\mathrm{d}[x_k^a]_r}{\sqrt{2\pi}}\right)\exp\left(\sum_{\substack{(i,j)\in\mathbb{L}\\1\leq a\leq b\leq n}}\sum_{r=1}^{N_i}\hat{q}_{i,j}^{a,b}[x_i^a]_r[x_j^b]_r\right). \tag{154}$$

Now setting $\phi(q,\hat{q}) = \Gamma(q,\hat{q}) + \Xi(q) + \psi(\hat{q})$, the saddle point method provides for $N$ large enough

$$\frac{\log J}{N} \simeq \mathrm{Extr}(\phi(q,\hat{q})). \tag{155}$$

**Replica symmetric ansatz.** Before computing the extremum we make the "replica symmteric ansatz": we assume for all $(i,j)$, $q_{i,j}^{a,b} = q_{i,j}\delta_{a,b}\delta_{(i,j)\in\mathbb{L}}$ and $\hat{q}_{i,j}^{a,b} = -\frac{1}{2}\hat{q}_{i,j}\delta_{a,b}\delta_{(i,j)\in\mathbb{L}}$. As shown here with this ansatz $\phi(q,\hat{q})$ will become tractable. We have

$$\Gamma(q,\hat{q}) = \frac{n}{2}\sum_{(i,j)\in\mathbb{L}} q_{i,j}\hat{q}_{i,j}. \tag{156}$$

Furthermore, we can calculate $\psi(\hat{q})$ noticing that

$$\psi(\hat{q}) = \frac{1}{N} \log \int_{\mathbb{R}^{n \times N}} \left( \prod_{\substack{1 \leq a \leq n \\ 1 \leq k \leq d}} \prod_{r=1}^{N_k} \frac{d[x_k^a]_r}{\sqrt{2\pi}} \right) \exp\left( -\frac{1}{2} \sum_{(i,j) \in \mathbb{L}} \hat{q}_{i,j} \sum_{r=1}^{N_i} \sum_{1 \leq a \leq n} [x_i^a]_r [x_j^a]_r \right), \quad (157)$$

so

$$\psi(\hat{q}) = \frac{n}{N} \log \int_{\mathbb{R}^N} \left( \prod_{1 \leq k \leq d} \prod_{r=1}^{N_k} \frac{d[x_k]_r}{\sqrt{2\pi}} \right) \exp\left( -\frac{1}{2} \sum_{(i,j) \in \mathbb{L}} \hat{q}_{i,j} \sum_{r=1}^{N_i} [x_i]_r [x_j]_r \right). \quad (158)$$

But notice also that we can group the terms. Defining the "equivalence class of $i$" as the set $\bar{i} = \{j | N_i = N_j\}$, we get

$$\psi(\hat{q}) = \frac{n}{N} \log \int_{\mathbb{R}^N} \prod_{1 \leq i \leq d} \prod_{r=1}^{N_i} \left( \frac{d[x_i]_r}{\sqrt{2\pi}} \prod_{j \in \bar{i}} e^{-\frac{1}{2}\hat{q}_{i,j}[x_i]_r[x_j]_r} \right). \quad (159)$$

But now, since the equivalence classes forms a partition $\mathcal{P}$ of $\{1, \ldots, d\}$, we have

$$\psi(\hat{q}) = \frac{n}{N} \log \int_{\mathbb{R}^N} \prod_{\bar{k} \in \mathcal{P}} \prod_{i \in \bar{k}} \prod_{r=1}^{N_k} \left( \frac{d[x_i]_r}{\sqrt{2\pi}} \prod_{j \in \bar{k}} e^{-\frac{1}{2}\hat{q}_{i,j}[x_i]_r[x_j]_r} \right). \quad (160)$$

Hence

$$\psi(\hat{q}) = \frac{n}{N} \log \prod_{\bar{k} \in \mathcal{P}} \left[ \int_{\mathbb{R}^{|\bar{k}|}} \prod_{i \in \bar{k}} \left( \frac{dy_i}{\sqrt{2\pi}} \prod_{j \in \bar{k}} e^{-\frac{1}{2}\hat{q}_{i,j}y_i y_j} \right) \right]^{N_k} \quad (161)$$

Or written in a slightly different way

$$\psi(\hat{q}) = n \sum_{\bar{k} \in \mathcal{P}} \frac{N_k}{N} \log \left[ \int_{\mathbb{R}^{|\bar{k}|}} \left( \prod_{i \in \bar{k}} \frac{dy_i}{\sqrt{2\pi}} \right) e^{-\frac{1}{2} \sum_{(i,j) \in \bar{k}} \hat{q}_{i,j}y_i y_j} \right]. \quad (162)$$

We define the overlap matrix $\hat{Q} = (\hat{q}_{i,j})$ and the sub-matrix $\hat{Q}^{\bar{k}} = (\hat{Q}_{i,j})_{(i,j) \in \bar{k}}$. Recalling that for a multivariate gaussian distribution

$$\int_{\mathbb{R}^{|\bar{k}|}} \left( \prod_{i \in \bar{k}} \frac{dy_i}{\sqrt{2\pi}} \right) e^{-\frac{1}{2} \sum_{(i,j) \in \bar{k}} \hat{q}_{i,j}y_i y_j} = \left( \det \hat{Q}^{\bar{k}} \right)^{-\frac{1}{2}} \quad (163)$$

we find

$$\psi(\hat{q}) = -\frac{n}{2} \sum_{\bar{k} \in \mathcal{P}} \frac{N_k}{N} \log \det \hat{Q}^{\bar{k}} \quad (164)$$

Finally, for the term $\Xi(q)$ we obtain

$$\Xi(q) = -\frac{n}{2} \sum_{(i,j) \in \mathbb{L}} z_{i,j} q_{i,j} + \frac{n}{4} \sum_{(i,j,k,l) \in \mathbb{T}} q_{i,k} q_{j,l} \sigma_{i,j}^{l,k}. \quad (165)$$

Summarizing, we have found

$$\phi(q, \hat{q}) = \frac{n}{2} \left\{ - \sum_{(i,j) \in \mathbb{L}} z_{i,j} q_{i,j} + \frac{1}{2} \sum_{\substack{i,j,k,l \\ \in \mathbb{T}}} q_{i,k} q_{j,l} \sigma_{i,j}^{l,k} + \sum_{(i,j) \in \mathbb{L}} q_{i,j} \hat{q}_{i,j} - \sum_{\bar{k} \in \mathcal{P}} \frac{N_k}{N} \log \det \hat{Q}^{\bar{k}} \right\}$$

$$\equiv \frac{n}{2} \tilde{\phi}(q, \hat{q}). \quad (166)$$

**Derivation of fixed point equation** (79). Now we will have to take derivatives to find the extremum of $\phi(q, \hat{q})$ or equivalently $\tilde{\phi}(q, \hat{q})$. In order to perform the derivatives it is useful to recall that for a symmetric matrix $X$ we have

$$\frac{\partial \log \det X}{\partial [X]_{i,j}} = \frac{1}{\det X} \frac{\partial \det X}{\partial [X]_{i,j}} = [X^{-1}]_{ji} = [X^{-1}]_{i,j}. \tag{167}$$

Therefore, we have for any $(i, j) \in \mathbb{L}$ (using the symmetry of $\sigma_{i,j}^{l,k}$)

$$\frac{\partial \tilde{\phi}(q, \hat{q})}{\partial \hat{q}_{i,j}} = 0 \implies q_{i,j} = \frac{N_i}{N} [(\hat{Q}^{\bar{i}})^{-1}]_{i,j} \tag{168}$$

$$\frac{\partial \tilde{\phi}(q, \hat{q})}{\partial q_{i,j}} = 0 \implies z_{i,j} = \hat{q}_{i,j} + \sum_{(k,l) \in \mathbb{L}} q_{k,l} \sigma_{i,k}^{l,j} \tag{169}$$

In matrix form, using the matrix $G$ with matrix elements $G_{i,j} = \frac{N}{N_i} Q_{i,j}$, and given an equivalence class $\bar{k}$ we have

$$Q^{\bar{k}} = \frac{N_k}{N} (\hat{Q}^{\bar{k}})^{-1} \tag{170}$$

$$B^{\bar{k}} = \hat{Q}^{\bar{k}} + \eta^{\bar{k}}(G) \tag{171}$$

where for any given matrix $D \in \mathbb{R}^{d \times d}$, the matrices $B^{\bar{k}}$, $\eta^{\bar{k}}(D)$ are the restriction of $B$, $\eta(D)$ on the subspace spanned by the basis $\mathcal{B}^{\bar{k}}$ (that is, on all the indices $(i, j) \in \bar{k} \times \bar{k}$), and $\eta(D)$ is defined such that for any $(i, j) \in \mathbb{L}$

$$[\eta(D)]_{i,j} = \sum_{(u,v) \in \mathbb{L}} \frac{N_u}{N} [D]_{u,v} \sigma_{i,u}^{v,j}. \tag{172}$$

So for any $k \in \{1, \ldots, d\}$ we have

$$B^{\bar{k}} = \left( \frac{N}{N_k} Q^{\bar{k}} \right)^{-1} + \eta^{\bar{k}}(G) \tag{173}$$

Hence using only $G$, because $G^{\bar{k}} = \frac{N}{N_k} Q^{\bar{k}}$, we obtain

$$B^{\bar{k}} G^{\bar{k}} = I_{|\bar{k}|} + \eta^{\bar{k}}(G) G^{\bar{k}}. \tag{174}$$

Notice now that $X \in \mathbb{R}^{|\bar{k}|} \mapsto \hat{Q}^{\bar{k}} X$ is related to the endomorphism restriction of $X \mapsto \hat{Q} X$ in the vector space spanned by the canonical basis $\mathcal{B}^{\bar{k}} = (e_i)_{i \in \bar{k}}$. In other words, we have that $\mathbb{R}^d = \bigoplus_{\bar{k} \in \mathcal{P}} \mathcal{B}^{\bar{k}}$, and that the subspaces $\mathcal{B}^{\bar{k}}$ are stable under action of $\hat{Q}$, but also under action of $B$ as there is also the constraint that $\forall (i, j) \notin \mathbb{L}, z_{i,j} = 0$. Similarly this is the case also for $\eta$, since by definition of $\eta$, we have $[\eta(D)]_{i,j} = 0$ for any $(i, j) \notin \mathbb{L}$. In other words, assuming the partition formed by the equivalence classes $\bar{i} = \{j | N_j = N_i\}$ is $\mathcal{P} = \{\bar{k}_1, \ldots, \bar{k}_p\}$, there exists a matrix $P \in \mathbb{R}^{d \times d}$ such that

$$P^{-1} B P = \left( \begin{array}{c|c|c|c} B^{\bar{k}_1} & 0 & \ldots & 0 \\ \hline 0 & B^{k_2} & \ldots & 0 \\ \hline \vdots & \vdots & \ddots & \vdots \\ \hline 0 & 0 & \ldots & B^{k_p} \end{array} \right), \quad P^{-1} G P = \left( \begin{array}{c|c|c|c} G^{\bar{k}_1} & 0 & \ldots & 0 \\ \hline 0 & G^{k_2} & \ldots & 0 \\ \hline \vdots & \vdots & \ddots & \vdots \\ \hline 0 & 0 & \ldots & G^{k_p} \end{array} \right) \tag{175}$$

and similarly with the same matrix $P$

$$P^{-1} \eta(D) P = \left( \begin{array}{c|c|c|c} \eta^{\bar{k}_1}(D) & 0 & \ldots & 0 \\ \hline 0 & \eta^{k_1}(D) & \ldots & 0 \\ \hline \vdots & \vdots & \ddots & \vdots \\ \hline 0 & 0 & \ldots & \eta^{k_p}(D) \end{array} \right) \tag{176}$$

Therefore, since we have for all $\bar{k} \in \mathcal{P}$ the equation (174), this is equivalent to having $(P^{-1} B P)(P^{-1} G P) = I_d + (P^{-1} \eta(D) P)(P^{-1} G P)$, in other words, this is equivalent to

$$BG = I_d + \eta(G) G \tag{177}$$

At this point we have derived the important fixed point equation (79).

**Derivation of equation** (78). Notice that we also have (because $M$ is symmetric)

$$\frac{\partial I}{\partial z_{i,j}} = \sum_{r,s=1}^{N} \mathbb{E} \frac{\partial [M]_{r,s}}{\partial z_{i,j}} \frac{\partial \log \det M}{\partial [M]_{r,s}} = \sum_{r,s=1}^{N} \mathbb{E} \frac{\partial [S]_{r,s}}{\partial z_{i,j}} [M^{-1}]_{r,s} = \mathbb{E} \text{Tr} \left[ (M^{-1})^{(i,j)} \right] \quad (178)$$

But on the other hand, with $(q^*, \hat{q}^*)$ the extrema of $\tilde{\phi}$

$$I = 2 \lim_{n \to 0} \frac{1 - \mathbb{E} J}{n} \simeq 2 \lim_{n \to 0} \frac{1 - e^{-N\frac{n}{2}\tilde{\phi}(q^*, \hat{q}^*)}}{n} = -N\tilde{\phi}(q^*, \hat{q}^*). \quad (179)$$

But notice that $\phi$ and $q, q^*$ are also themselves functions of $B$, in other words

$$\frac{I}{N} \simeq -\tilde{\phi}(q^*(B), \hat{q}^*(B), B), \quad (180)$$

and hence using chain rule, and remembering that $\frac{\partial \tilde{\phi}}{\partial q_{i,j}} = \frac{\partial \tilde{\phi}}{\partial \hat{q}_{i,j}} = 0$ in $q^*, \hat{q}^*$, we have

$$\frac{1}{N} \frac{\partial I}{\partial z_{i,j}} = -\sum_{r,s=1}^{d} \frac{\partial [B]_{r,s}}{\partial z_{i,j}} \frac{\partial \tilde{\phi}}{\partial [B]_{r,s}} (q^*, \hat{q}^*, B) + 0 + 0 = q_{i,j}^* = \frac{N_i}{N} [G^*]_{i,j}. \quad (181)$$

Eventually we obtain

$$[G^*]_{i,j} = \lim_{N \to \infty} \frac{1}{N_i} \mathbb{E} \text{Tr} \left[ (M^{-1})^{(i,j)} \right] \quad (182)$$

which is (78).

**Derivation of equation** (81). Regarding $\sigma_{i,j}^{l,k}$, notice that we can express the random block $R^{(i,j)}$ in the following way

$$R^{(i,j)} = \sum_{t \in \mathbb{S}_{(i,j)}} \alpha_{i,j}^t Y_t + \sum_{t \in \mathbb{S}_{(j,i)}} \alpha_{j,i}^t Y_t^{\mathsf{T}} \quad (183)$$

so, provided $r, s$ are chosen such that $r \neq s$, we find

$$N \cdot \mathbb{E} \left[ [R^{(i,j)}]_{r,s} [R^{(l,k)}]_{s,r} \right] = \sum_{t \in \mathbb{S}_{(i,j)}} \alpha_{i,j}^t \alpha_{k,l}^t + \sum_{t \in \mathbb{S}_{(j,i)}} \alpha_{j,i}^t \alpha_{l,k}^t = \sigma_{i,j}^{l,k} \quad (184)$$

which is nothing else than (81).

**A note on correlated random matrices.** To extend further the result, notice that we can always construct standard gaussian random blocks, say $R^{(i,j)}$ and $R^{(l,k)}$, such that they have a priori some covariance $v$ with $v = \mathbb{E} \left[ [R^{(i,j)}]_{r,s} [R^{(l,k)}]_{s,r} \right]$. While we stated a result where these blocks are built from a sum of $(Y_k)$ which are standard gaussian random matrices, notice that it is always possible to use two independent standard random matrices $Y_1, Y_2$, and define: $R^{(i,j)} = vY_1 + \sqrt{1 - v^2} Y_2$ and $R^{(k,l)} = Y_1^T$. Therefore, the result remains valid even in the general case where we only suppose that the blocks in $R$ are distributed following a gaussian distribution, with potentially some entry-wise covariance and using equation (184) as the definition of $\sigma_{i,j}^{l,k}$.

## F Numerical results

All the experiments are run on a standard desktop configuration:

1. Matlab R2019b is used to generate the heatmaps or 3D landscapes. Most exemples can be generated in less than 12h on a standard machine.

2. The experimental comparisons run on a standard instance of a Google collaboratory notebook in less than a few hours.

## F.1 Numerical computations

We take equation (4) as an example of how to proceed with the numerical experiments. Specifically we consider the second integral in the Cauchy integral representation of $g(t)$

$$g_2(t) = -\frac{1}{2i\pi} \oint_\Gamma dz \left\{ \frac{1 - e^{-t(z+\delta)}}{z+\delta} K(z) \right\}. \tag{185}$$

We choose a contour with $\lambda^* = \max \text{Sp} \frac{Z^T Z}{N}$ with two positive fixed constants $\epsilon, \Delta$:

$$\Gamma = \{\gamma\lambda^* \pm i\Delta | -\epsilon \le \gamma \le 1 + \epsilon\} \cup \{\epsilon\lambda^* + \gamma i\Delta | -1 \le \gamma \le 1\} \cup \{-\epsilon + \gamma i\Delta | -1 \le \gamma \le 1\} \tag{186}$$

Now, the integrand is continuous in $\lambda^* + \epsilon$ and $-\epsilon$ for $\epsilon$ small enough. So taking the limit $\epsilon \to 0$ and $\Delta \to 0$

$$g(t) = \lim_{\Delta \to 0} \frac{1}{2i\pi} \int_0^{\lambda^*} \left\{ \frac{1 - e^{-t(r+\delta+i\Delta)}}{r+\delta+i\Delta} K(r+i\Delta) - \frac{1 - e^{-t(r+\delta)-i\Delta}}{r+\delta-i\Delta} K(r-i\Delta) \right\} dr \tag{187}$$

which is simply

$$g(t) = \int_0^{\lambda^*} \frac{1 - e^{-t(r+\delta)}}{r+\delta} \lim_{\Delta \to 0} \frac{1}{2i\pi} \left\{ K(r+i\Delta) - K(r-i\Delta) \right\} dr \tag{188}$$

Obviously the inward term is also given by the limit $\lim_{\Delta \to 0} \frac{1}{\pi} \text{Im} K(r + i\Delta)$. So this all there is to compute from the former algebraic equations are appropriate imaginary parts. This can be done by taking a discretized interval $0 \le r_1 \le \ldots \le r_K \le \lambda^*$, and solving the algebraic equations for the imaginary value $\text{Im} t_1^x$ for $x = r_i$, $i = 1, \cdots, K$.

We proceed similarly with the terms containing two complex variables $x$ and $y$ (or two resolvents). For instance for $W(x, y)$ one uses the limit in $\Delta_x, \Delta_y \to 0$ of $\rho(x, y)$ where

$$\rho(x, y) = \lim_{\Delta_x \to 0} \lim_{\Delta_y \to 0} \left[ \frac{-1}{4\pi^2} \left\{ W(r_x + i\Delta_x, r_y + i\Delta_y) - W(r_x + i\Delta_x, r_y - i\Delta_y) \right\} \right.$$
$$\left. - \frac{-1}{4\pi^2} \left\{ W(r_x - i\Delta_x, r_y + i\Delta_y) - W(r_x - i\Delta_x, r_y - i\Delta_y) \right\} \right] \tag{189}$$

or equivalenlty

$$\rho(x, y) = \lim_{\Delta_x, \Delta_y \to 0} \frac{1}{2\pi^2} \text{Re} \left\{ W(r_x + i\Delta_x, r_y - i\Delta_y) - W(r_x + i\Delta_x, r_y + i\Delta_y) \right\} \tag{190}$$

## F.2 Technical considerations

**Dirac distributions with 1-variable functions:** It happens that the limiting distribution $\frac{Z^T Z}{N}$ may contain a mixture of a Dirac peak at 0 and a continuous measure. For instance, $K(z)$ may contain a branch cut in the interval $\mathcal{C}^* = [\lambda_1, \lambda^*]$ with $\lambda_0 = 0 < \lambda_1 < \lambda^* < \infty$ along with an isolated pole in 0 with: $K(z) = \frac{\alpha}{0-z} + K_c(z)$ (where $K_c : \mathbb{C} \setminus \mathcal{C}^* \to \mathbb{C}$). For instance, equation (188) becomes:

$$g(t) = \alpha \frac{1 - e^{-t\delta}}{\delta} + \int_{\lambda_0}^{\lambda^*} dr \frac{1 - e^{-t(\delta+r)}}{r+\delta} \lim_{\Delta \to 0} \frac{1}{\pi} \text{Im} K_c(r+i\Delta) \tag{191}$$

The weight $\alpha$ can be retrieved by computing $\alpha = \lim_{\epsilon \to 0^+} (-i\epsilon) K(i\epsilon) = \lim_{\epsilon \to 0^+} \epsilon \text{Im} K(i\epsilon)$.

**Dirac distributions with 2-variables functions:** Similarly, we can have an isolated pole at 0 for $x, y$ for $W(x, y)$. In that case, we can write down $W(x, y)$ as for instance:

$$W(x, y) = \frac{\alpha_{xy}}{(0-x)(0-y)} + \frac{\alpha_x}{0-x} W_y(y) + \frac{\alpha_y}{0-y} W_x(x) + W_{xy}(x, y) \tag{192}$$

where $W_x, W_y$ are defined on $\mathbb{C} \setminus \mathcal{C}^* \to \mathbb{C}$ and $W_{xy} : (\mathbb{C} \setminus \mathcal{C}^*)^2 \to \mathbb{C}$. Firstly, We can easily find $\alpha_{xy}$ with:

$$\alpha_{xy} = \lim_{\epsilon \to 0^+} (-\epsilon^2) \text{Re} W(i\epsilon, i\epsilon) \tag{193}$$

Secondly, all the considered 2-variables functions are symmetrical with respect to $x$ and $y$: $W(x, y) = W(y, x)$ which implies that $\alpha_x = \alpha_y$ and $W_x(r) = W_y(r)$ for all $r \in \mathbb{C} \setminus \mathcal{C}^*$. Therefore, if we have $\gamma_t(z) = \frac{1 - e^{-t(\delta + z)}}{z + \delta}$, we have to compute:

$$\mathcal{R}_{x,y}\{\gamma_t(x)\gamma_t(y)W(x,y)\} = \gamma_t(0)^2\alpha_{xy} + \iint_{[\lambda_0, \lambda^*]^2} \gamma_t(u)\gamma_t(v)\rho(u,v)\mathrm{d}u\mathrm{d}v$$
$$+ 2\gamma_t(0)\int_{\lambda_0}^{\lambda^*} \mathrm{d}r\gamma_t(r)\lim_{\Delta \to 0^+}\frac{\alpha_x}{2i\pi}\left\{W_y(r + i\Delta) - W_y(r - i\Delta)\right\} \tag{194}$$

But because we don't have access to $\alpha_x$ nor $W_y$ directly, we can use the full form:

$$\mathcal{R}_{x,y}\{\gamma_t(x)\gamma_t(y)W(x,y)\} = \gamma_t(0)^2\alpha_{xy} + \iint_{[\lambda_0, \lambda^*]^2} \gamma_t(u)\gamma_t(v)\rho(u,v)\mathrm{d}u\mathrm{d}v$$
$$+ 2\gamma_t(0)\int_{\lambda_0}^{\lambda^*} \mathrm{d}r\gamma_t(r)\lim_{\Delta \to 0^+}\lim_{\epsilon \to 0^+}\frac{-i\epsilon}{2i\pi}\left\{W(i\epsilon, r + i\Delta) - W(i\epsilon, r - i\Delta)\right\} \tag{195}$$

This comes from the fact that for $\epsilon \to 0$ we have: $W(i\epsilon, r + i\Delta) \sim \frac{\alpha_x}{-i\epsilon}W_y(r + i\Delta)$. Because we expect a real result, we ought to have numerically:

$$\mathcal{R}_{x,y}\{\gamma_t(x)\gamma_t(y)W(x,y)\} = \gamma_t(0)^2\alpha_{xy} + \iint_{[\lambda_0, \lambda^*]^2} \gamma_t(u)\gamma_t(v)\rho(u,v)\mathrm{d}u\mathrm{d}v$$
$$+ \gamma_t(0)\int_{\lambda_0}^{\lambda^*} \mathrm{d}r\gamma_t(r)\lim_{\Delta \to 0^+}\lim_{\epsilon \to 0^+}\frac{\epsilon}{\pi}\mathrm{Re}\left\{W(i\epsilon, r - i\Delta) - W(i\epsilon, r + i\Delta)\right\} \tag{196}$$

**1-variable distributions in 2-variables functions** Finally, it can happen that the 2-variables functions $W(x, y)$ actually generates a distribution $\rho(u, v) = \rho_c(u, v) + \mu(u)\delta(v - u)$ which may be the sum of a continuous measure $\rho_c(u, v)$ as described above, and another measure $\mu(u)\delta(v - u) = \delta(u - v)\mu(v)$.

## F.3 Additional heatmaps

We provide additional heatmaps that complement those of Sect. 3. Notice that all the heat-maps are always derived from a 3D mesh comprising $30 \times 100$ points as in Fig. 12.

Instead of fixing $\lambda$, we can rescale it and fix $\delta = c\lambda$. As we have seen, the $\lambda$ parameter seems to affect the length of the time scale on the first plateau. Rescaling it as seen in Fig. 6, the interpolation threshold time scale becomes constant in the over-parametrized regime at fixed $\delta$, and the results are consistent with what is observed empirically in [22].

We notice also that under the configuration in Fig. 7 where $r = 0$ (the noise of the second layer vanishes), the second plateau seems to vanish with the test error.

One of the effects of a large $\lambda$ is that it removes the double descent on the test error, which is consistent with the description in [17]. Another effect is that it seems to add an additional "two-stage decrease" in the training error as can be seen in Fig. 9 and also in the experiments in Figs. 15, 16.

Note that the previous figures are perfomed for the activation function $\sigma(x) = \mathrm{Relu}(x) - \frac{1}{\sqrt{2\pi}}$ while Figs. 10 and 11 are displayed other activation functions, $\sigma(x) = \tanh(x)$ and $\sigma(x) = \tanh(5x)$. We can see that the epoch-wise structures are more marked when the slope of the activation function is bigger in the second case.

## F.4 Comparison with experimental simulations

We have already shown on figure 2 in Sect. 3 that the analytical formulas for the training and generalization errors match the experimental curves in the limit of $t \to +\infty$. Here we provide additional evidence that this is also the case for the whole time-evolution in Figs. 13 and 14 as the dimension $d$ increases.

In 15, 16, we can see that the epoch-wise descent structures of the training error and test error can be captured correctly experimentally for long time. Note that we have taken $d = 100$ small enough to be able to run these experiments for such a long timescale.

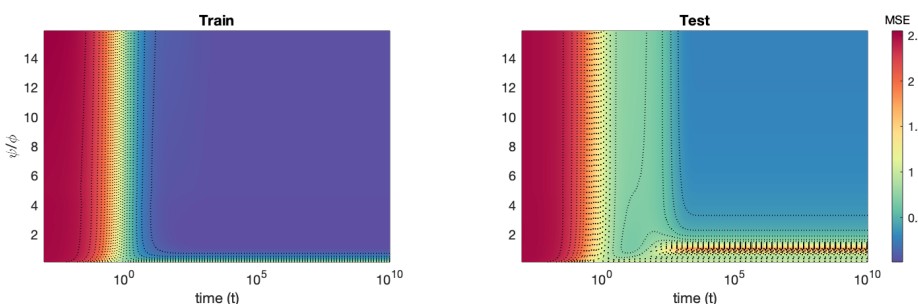

Figure 6: Analytical training error and test error evolution at fixed $\delta$ with parameters $(\mu, \nu, \phi, r, s, \delta) = (0.5, 0.3, 3, 2., 0.4, 0.001)$

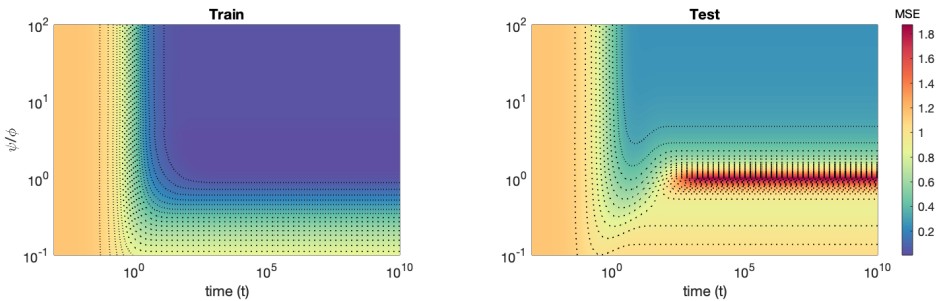

Figure 7: Analytical training error and test error evolution with parameters $(\mu, \nu, \phi, r, s, \lambda) = (0.5, 0.3, 3, 0, 0.4, 0.001)$

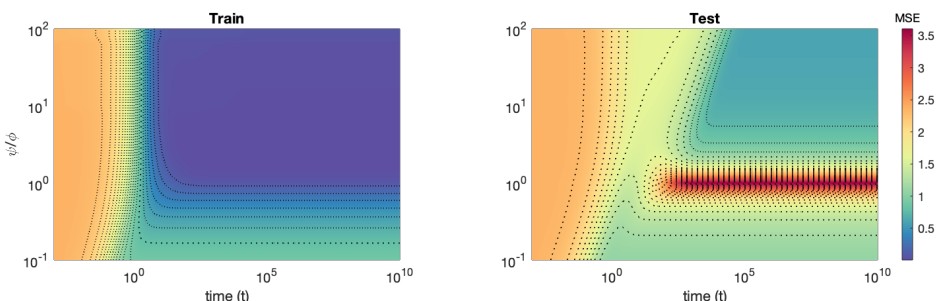

Figure 8: Analytical training error and test error evolution with parameters $(\mu, \nu, \phi, r, s, \lambda) = (0.5, 0.3, 0.5, 2, 0.1, 0.003)$

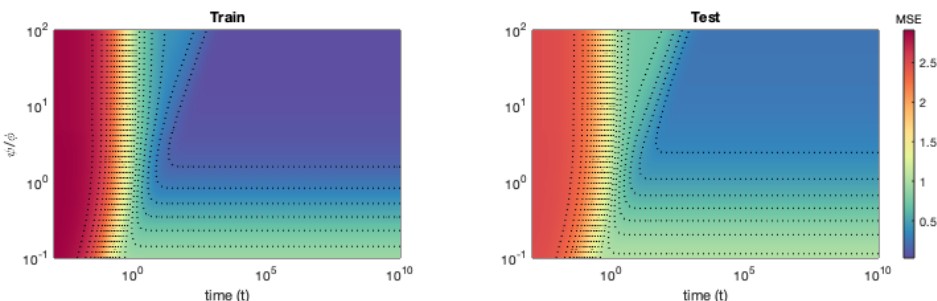

Figure 9: Analytical training error and test error evolution with parameters $(\mu, \nu, \phi, r, s, \lambda) = (0.5, 0.3, 3, 0, 0.4, 0.1)$

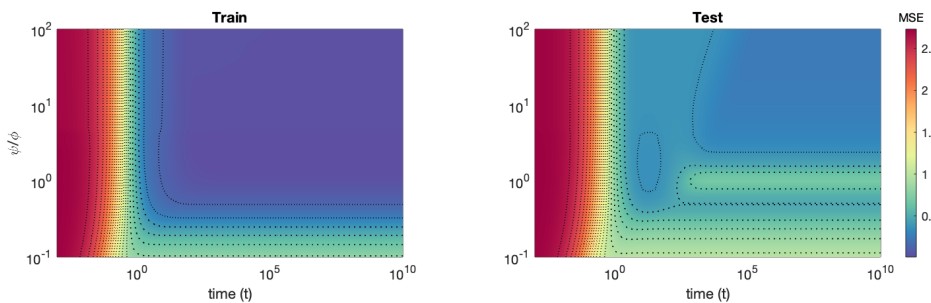

Figure 10: Analytical training error and test error evolution with parameters corresponding to $\sigma(x) = \tanh(x)$ with $(\mu, \nu, \phi, r, s, \lambda) = (0.61, 0.15, 3, 0, 0.4, 0.001)$

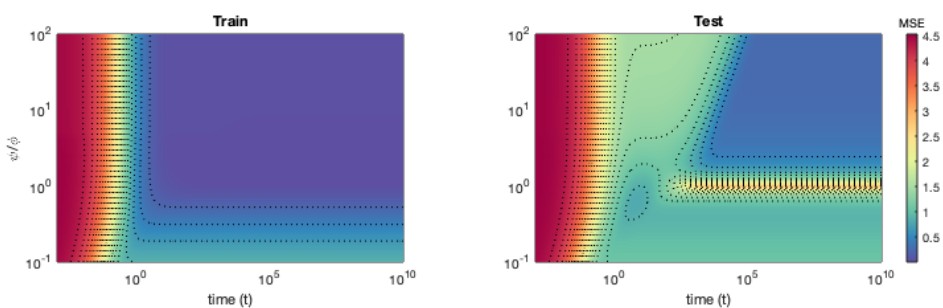

Figure 11: Analytical training error and test error evolution with parameters corresponding to $\sigma(x) = \tanh(5x)$ with $(\mu, \nu, \phi, r, s, \lambda) = (0.79, 0.47, 3, 2, 0.4, 0.001)$

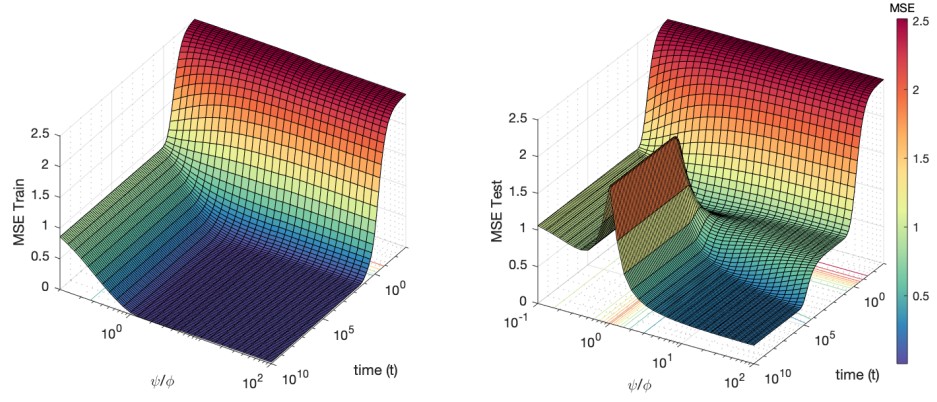

Figure 12: Analytical training error with parameters $(\mu, \nu, \phi, r, s, \lambda) = (0.5, 0.3, 3, 2., 0.4, 0.001)$

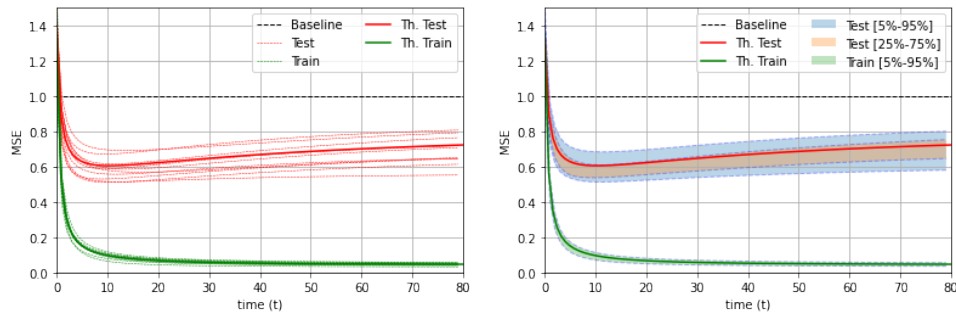

Figure 13: Analytical training error and test error profile with parameters $(\mu, \nu, \phi, \psi, r, s, \lambda) = (0.5, 0.3014, 1.4, 1.8, 1.0, 0, 0.01)$ compared to 10 experimental runs $(\sigma = \mathrm{Relu} - \frac{1}{\sqrt{2\pi}})$ with $d = 200$ and $\mathrm{d}t = 0.01$

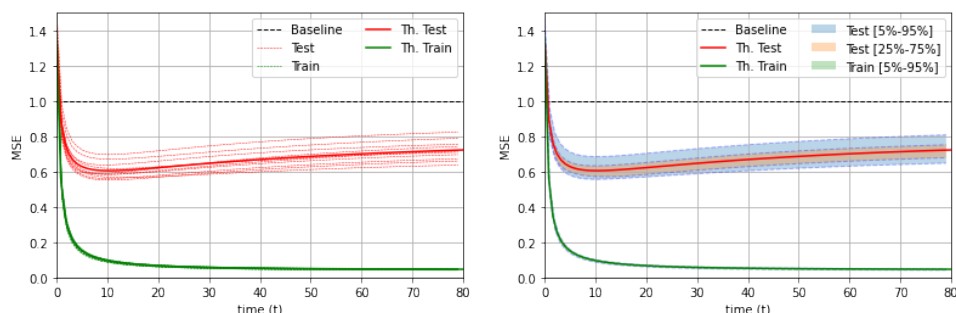

Figure 14: Analytical training error and test error profile with parameters $(\mu, \nu, \phi, \psi, r, s, \lambda) = (0.5, 0.3014, 1.4, 1.8, 1.0, 0, 0.01)$ compared to 10 experimental runs $(\sigma = \mathrm{Relu} - \frac{1}{\sqrt{2\pi}})$ with $d = 1000$ and $\mathrm{d}t = 0.01$

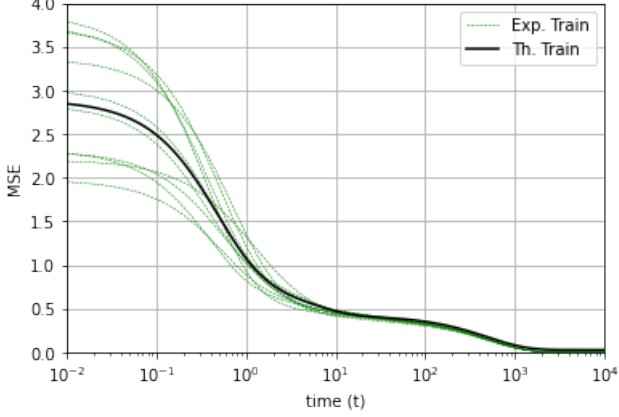

Figure 15: Analytical training error with parameters $(\mu, \nu, \phi, \psi, r, s, \lambda) = (0.5, 0.3, 300, 3, 2, 0.4, 0.1)$ compared to 10 experimental runs $(\sigma = \mathrm{Relu} - \frac{1}{\sqrt{2\pi}})$ with $d = 100$ and $\mathrm{d}t = 0.01$

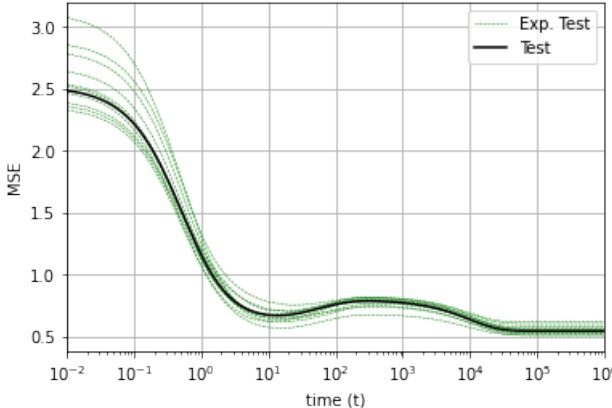

Figure 16: Analytical test error with parameters $(\mu, \nu, \phi, \psi, r, s, \lambda) = (0.5, 0.3, 6, 3, 2, 0.4, 0.0001)$ compared to 10 experimental runs with $d = 100$ and $\mathrm{d}t = 0.01$ for $0 \leq t \leq 10^4$ and $\mathrm{d}t = 0.1$ for $10^4 \leq t \leq 10^6$