# OpenReview forum: "Model, sample, and epoch-wise descents: exact solution of gradient flow in the random feature model"
_NeurIPS.cc/2021/Conference — NeurIPS 2021 Poster_

### Official Review · Reviewer_wJHk · 2021-07-06

**Rating:** 7
**Confidence:** 4

**Summary:**

This paper presents an analytical derivation of the train and test loss curves of the random feature model. The main novelty lies in the fact that the authors not only capture the behavior at final times, but also the dynamics. This leads to various interesting observations, since such models have been shown to exhibit epoch-wise double descent.


**Limitations And Societal Impact:**

Yes

**Main Review:**

Overall, I recommend the acceptance of this paper to NeurIPS.

Strengths:
- First analysis of epoch-wise double descent I know of for RF models
- Paper is well organised, results are presented clearly
- The authors clearly state the non-rigorous hypothesis used
- The phase space plots are informative and reveal new phenomena such as the exponentially elongated bump in the epoch-wise phase space as we regularise too little (this is particularly interesting in my opinion)
- The authors do a good job to provide a sketch of proof, and even include an alternative proof using a different technique from statistical physics
- The derivations in the SM are thorough and detailed

Weaknesses:
- Could be viewed as slightly incremental to the non-initiated community, but the complexity of the analytical derivations justify the novelty of the paper
- Fig. 2, which overlays the theoretical and experimental results, is plotted at final time as a function of phi/psi, but the late-time error was already derived in previous works; it would be better to have the numerical confirmation as a function of time, since time-dependency which is the key novelty of this paper.
- I think more space should be devoted to the phenomenology observed, as the analytical results are nearly impossible to parse for the reader.

Nits:
- Fig 1: a bit confusing that the t axis are inverted in the two panels
- Line 114: Hermite polynomials should be indexed by n
- “we notice that the linear-spike seems to appear earlier than the non-linear one” : this was observed already in reference 21.


**Time Spent Reviewing:**

3

---

> ### Author Response · Authors · 2021-08-09
> **Answer to Reviewer wJHk**
>
> Dear Reviewer, thank you for your remarks and for the last minor remarks which will help us improve the paper further.
>
> As this is also pointed out by reviewer b8CN, the analytical result is indeed quite complex to parse at first glance and more discussion in sect 3.2 would be welcome. We agree that the analytical formula seems difficult to interpret in the first place. Nevertheless, analytical formulas are also always desirable to better understand paradigms, and it is the application of these formulas that allow us to develop an intuition. We already learn from their application that the model has a rich and non-trivial phenomenology (and we believe that this was not widely anticipated). We will do our best to amplify the discussion of sect. 3.2 as stated in the general comment.

---

### Official Review · Reviewer_b8CN · 2021-07-16

**Rating:** 7
**Confidence:** 4

**Summary:**

This article provides, modulo some important technical assumptions, computationally tractable formulas for the test and train error of a random features model trained by gradient flow on \ell_2-regularized MSE during the entire course of training. The features are those computed by the first layer of a randomly initialized neural network and the regime of interest for this article is a triple scaling limit in which the input dimension, dataset size, and hidden layer width tend to infinity but their ratios remain order 1. Unlike prior work this article provides formulas for the evolution in time of both the training and test error, instead only of only their behaviors are t = \infty. While the results are not completely rigorous (they assume concentration for spectral statistics of certain random matrices), I found them intriguing and am fairly convinced by plots in which the predictions fit empirics quite well. The main weakness for me is a lack of further formulation of takeaways from the theoretical results (see weakness #2). Overall, I thought this was a solid paper.

**Limitations And Societal Impact:**

The limitations were discussed. Societal impact was not addressed, but it is not clear to me that this kind of article has any serious potential societal impacts.

**Main Review:**

Strengths:

1. Understanding the regime where input dimension, dataset size and the number of model parameters are all large is clearly relevant to modern ML.

2. The techniques in this article give what is, at least to me, a nice approach to understanding the behavior in time for train and test error in random feature models. While these models are linear in the trainable parameters, they are not linear in the inputs and understanding the resulting training curves seems hard and interesting.

3. If I understood correctly, the plots in Figure 3,4,5 were generated by analytically computing the test and train error from the formulas in Result 3.1. Assuming these match empirics, this is very useful indeed. I particularly liked “seeing” the epoch-wise, model-wise, and sample-wise descents appear and the phase transitions between double descent, triple descent, and other behaviors.

4. I thought the main body of the article was well-written and was able to follow the logic.

Weaknesses:
1. The formulas in Result 3.1, while seemingly analytically computable, are sort of horrendous at first glance. It would be nice if some intuition could be provided for what is going on.

2. An amplification of #1: I feel that the techniques in this article are sophisticated but the interpretation of the results is a bit lackluster. Concretely, since the formulas themselves are hard to grok, it would have been very useful to me as a reader if the discussion in section 3.2 were amplified. (I understand that space is an issue, but I am not sure anyone really needs to see the exact structure of the linear pencil on page 9?) For example:

2a. When exactly do we expect that early stopping is useful? Are any of these situations surprising?
2b. I am intrigued by the double plateau. Can you explain it intuitively? When should one expect it to occur?
2c. Can you explain more about the interaction of the non-linearity via its first few Hermite coefficients and the simultaneous appearance of the linear and non-linear spike?


Minor:
1. “passed an interpolation” should probably be “past an interpolation”

2. For equation (1) in the definition of H^{train} either H^{train} should be evaluated at a instead of a_t or the right hand side should contain ||a_t||^2 instead of ||a||^2.

3. Line 179 page 5 “time” should probably be “times”

4. It might be useful for the reader if the extra empirical validation of the results in Figures 13-15 were highlighted in the main body.

5. I am not an expert in the developments of this kind of high-dimensional RMT for random feature generalization, but I have the feeling that the article of Adlam-Pennington “The neural tangent kernel … “ is kind of given short shrift. Maybe I’m missing the point, but it seems that the use of linear pencils is key for doing the computations related to spectral curves and the algebraic relations in Result 3.1. And if I am not mistaken these techniques were used for first time in this context in that article. More broadly, as a non-expert, it would have been very useful to have a bit more context comparing the present work to the various different strands of work on double and triple descent in high dimensional models.

**Time Spent Reviewing:**

3 hours

---

> ### Author Response · Authors · 2021-08-09
> **Answer to Reviewer b8CN**
>
> Dear reviewer, thank you for your review and for the last remarks which will help us improve the paper further.
>
> Concerning a remark/suggestion about space we still want to try and keep the linear pencil on page 9 since it is a crucial part of the technique. We also believe that this allows the readers to better follow the logic of the analysis in the supplementary material. To the best of our knowledge the more systematic use of linear pencils in RMT and the fixed point equations stemming from them goes back to refs 28, 29 (reviewed in 35). But we agree that the very recent work of Adlam and Pennington (which we cite related to the gaussian equivalence principle) should be cited more prominently.
>
> Concerning early stopping (2a) it seems that it is useful essentially in regimes when psi and phi are of the same order. This is maybe not too surprising when one already knows that in such regimes there are double or triple descent peaks at infinite time. However, now, here we clearly learn that these peaks develop suddenly time-wise: this is a rather sharp crossover (not to say a "phase transition") as a function of time.  We admittedly do not have a clear intuition to offer for why the double plateau appears (2b) but we tend to observe it in very highly overparameterized regimes (large ratio psi/phi). Concerning question 2c this kind of phenomena has been discussed in ref 21 (as also pointed out by reviewer wJHk - thanks - we should point this out indeed) and depending on space we will comment more on the role of the structure of the non-linearity.

---

### Official Review · Reviewer_U78k · 2021-07-17

**Rating:** 8
**Confidence:** 3

**Summary:**

The authors analytically study model, sample, and epochwise descents of the random feature model. Model descent meaning how the test error changes as the model complexity is varied; sample descent meaning how the test error changes as the number of samples are varied; and epochwise descent meaning how the test error changes as the model trains. They derive analytic expressions for the test and train errors and study what happens when the number of neurons $N$, the number of samples $n$, and the sample dimensionality $d$ all tend to infinity. Since this can be done in varying ways, they study when $\frac{N}{d} \rightarrow \psi$ and $\frac{n}{d} \rightarrow \phi$ for fixed values of $\psi$ and $\phi$. They observe double descent which is beneficial (lower test error when $\psi \gg \phi$) if training is performed for a long time. On the other hand, if training is cut short (early stopping) the test error is low and even has a dip where later there develops a peak. The authors also observe triple descent if the training time tends to $\infty$.

**Limitations And Societal Impact:**

Limitations:
As the authors mention on line 88, the work is not completely rigorous because of their concentration hypothesis. However, this is not an issue for me as they provide supporting empirical evidence to support their work.

Potential negative societal impacts:
None

**Main Review:**

Originality:

The idea of studying the analytic expressions for test and train error as training evolves is not particularly novel. However, the authors use some very sophisticated mathematical techniques which bring novelty to the work in a way that is not typical. Good job.

Quality:

This paper is extremely well written and flushed out. The authors did a great job.

Clarity:

The paper is mostly clear. It is hard to read because of the notation; however, that is to be expected for such a theoretical paper.

Significance:

This work is very interesting and important for the continued understanding of more complex models such as deeper neural networks.

**Time Spent Reviewing:**

7

---

> ### Author Response · Authors · 2021-08-09
> **Answer to Reviewer U78k**
>
> Dear reviewer, thank you for your comment!

---

### Official Review · Reviewer_Crju · 2021-07-19

**Rating:** 7
**Confidence:** 2

**Summary:**

In this paper, authors proposed an analytical model to understand the multiple-descent curves in generalization errors of over-parameterized models (random feature models optimized over gradient flow). More specifically, they provided exact solutions to interpret the model-wise, epoch-wise and sample-wise double descent phenomenon.

In Section 2., authors define the learning problem of over-parameterized models with random features as a regression problem of Ridge-alike, then define the optimization procedure as an ODE. In Section 2.2., authors represent  training and testing errors with  Cauchy integral representations with assumptions prespecified. They report the result as the solution of ODE under assumptions while providing some simulation results to backup their theoretical results. The sketched proofs are attached.

**Ethics Review Area:**

["I don’t know"]

**Limitations And Societal Impact:**

It helps us understand why over-parameterized models work and interpret their generalization performance.

**Main Review:**

The work provides asymptotic results on the generalization performance I might be wrong, the main results of this work looks the combination of existing results on gradient flows (to run OLS or Ridge) with some results on random matrix theories (to model the sample covariance established by random features) via Cauchy integral. Authors are encouraged to discuss their results with "Just Interpolate: Kernel ''Ridgeless'' Regression Can Generalize" at Annals of Stats 2020. Of-course, in Ridgeless paper, they did not cover the optimization issue. With gradient flow for OLS and Cauchy integral, it seems possible to obtain the exact solution of or tight approximation to the gradient flow for OLS at continuous-time t. It looks authors obtained their results through combination. Some simulations could be improved using real-world datasets.

**Time Spent Reviewing:**

5hrs

---

> ### Author Response · Authors · 2021-08-09
> **Answer to Reviewer Crju**
>
> Dear reviewer, thank you first for the additional reference which will be included and discussed in a newer revision.
>
> One of the main objectives of the paper is to demonstrate that the similar phenomena unraveled from the experiments of Nakkiran, Kaplun et Al. (2019) [22] can be obtained with an analytical result on a random features model. We agree that exploring our results further with a real-world dataset is a tempting suggestion but we would reserve this for a parallel work, since the theoretical analysis presented here is already significantly non-trivial and long.

---

> > ### Comment · Reviewer_Crju · 2021-08-25
> > **Thanks for your response.**
> >
> > I believe authors' responses have clearly addressed my concerns.

---

### Author Response · Authors · 2021-08-09
**General Answer**

Dear reviewers,

We thank you all, for your time and commitment reading the paper, as well as your constructive comments, that we will take into account in the revised version.

As a general comment, we will do our best to amplify the discussion of section 3.2:

-- One lesson is the structural form of the full time dependence: it is given by spectral densities whose Stieltjes transforms are solutions of "algebraic only" equations. This structure could well be generic and serve as a guide for ansatze in other or more complicated situations. In our opinion the structure could also be interesting to mathematicians.

-- Other lessons are obtained through the numerical plots that these analytical formulas yield along the various dimensions of time, model parameters and regularization. Besides double and triple descents, we observe a double plateau structure as well as an elongated bump structure for very small regularization.
All these points certainly deserve more investigation in order to better understand their origin and we hope that this work can spur more research.

As a last comment: we think the present paper is important because it shows a novel method using a highly non-trivial combination of existing tools to analyze a learning problem. As an example, we are currently applying the same method to multiple fixed-layers to gain a better understanding of the generic structures that appear in the generalization error.

---

### Comment · Reviewer_b8CN · 2021-09-02
**Good paper, but I feel prior work is not appropriately credited**

As a reviewer of this article, I have read the authors' response and again looked at the article as well as some publicly available material. I still like the article but have come to be even less happy than before about the credit given to prior work using linear pencils to understand random feature regression (I mentioned this point in my original review).

Specifically, it seems to me that the seminal paper that brought the idea of using linear pencils and operator valued free probability (OVFP) to studying random features was the work [32] of Adlam and Pennington. I understand that this paper didn't invent linear pencils and OVFP but it both pointed out the non-trivial fact that they are useful in the context of high dimensional random feature models and some did extremely precise and complicated computations to understand the generalization error (among other things). I believe that this article  should be highlighted much more prominently.

I feel this way in part because I saw that obtaining a Cauchy integral representation for the time-varying training/test loss appears in several talks by the authors of [32] (one example is around 15:50 in this video: https://www.youtube.com/watch?v=kujyufpr9ys&list=PLWQvhvMdDChzsThHFe4lYAff3pu2m0v2H&index=8 see also around 40:00 in that video for plot of loss vs. time). Of course I don't expect the authors of the article under review to have watched all possible videos on the subject. But since what I view as the key new idea in the article under review (namely the time-varying expression for the training/test error) was already discussed in a well-attended public forum, I think it should be explicitly mentioned. From my point of view, giving accurate attribution for where both ideas and techniques come from is important for the integrity of our field.

I hope the authors will consider making this emendation, as I feel it is both fair to do so and helpful for giving prospective readers more context.

---

> ### Author Response · Authors · 2021-09-03
> **Response to official comment of reviewer b8CN**
>
> As briefly promised in our first response, we are committed to fairly and better cite the paper of Adlam & Pennington [32] in relation with the introduction of linear-pencils in the context of machine learning. This will be done in the introduction in the paragraph related to the techniques used in this work. This will also be reminded to the reader in section 4.3 and corresponding SM. We thank the referee for pointing us towards the link of the interesting talk (which was unknown to us) and we will duly acknowledge and credit this video also.
>
> In our paper, the analysis of the time dependence of the generalization error is fully carried out in all details - something that we have not found in the literature to the best of our knowledge. Also we provide complete formulas in results 3.1, 3.2. Furthermore, we compute numerically in detail the predictions of these asymptotic formulas and compare them with simulations for finite size models and give error bars. Besides, an additional contribution is the derivation of the fixed-point equation through the replica method. This derivation is different from the one found in the literature through free-probability theory.

---

### Decision · Program_Chairs · 2021-09-27

**Decision:**

Accept (Poster)

**Comment:**

This paper presents an asymptotically exact formula for the time-dependence of the training and test error of random feature regression in high dimensions. The reviewers all maintain a favorable view of the results, but there was some concern about the credit afforded to prior work.

Indeed, operator-valued free probability and linear pencils (and the linearizations required to utilize them in the context of nonlinear random feature models) were introduced to the machine learning community in prior work with the express purpose of facilitating the kind of analysis undertaken here. Furthermore, it appears that the main results and methods of this paper, namely those derived via the Cauchy integral formula, have been obtained and advertised elsewhere (though apparently only in talks and never published). As such, it seems there is limited technical novelty in the current paper, and it is imperative that the discussion is modified to more properly reflect this prior work and to emphasize that the contributions of this paper are not the techniques themselves, but rather the results.

On the other hand, there is of course significant value in actually carrying through the calculations in detail, presenting concrete results, and analyzing their implications. As such, this paper will be of value to the community, irrespective of related prior work. Therefore I believe this paper is a good addition to NeurIPS and recommend acceptance.